# Policy Guided Tree Search for Enhanced LLM Reasoning

**Yang Li** [1]

## Abstract

Despite their remarkable capabilities, large language models often struggle with tasks requiring complex reasoning and planning. While existing approaches like Chain-of-Thought prompting and tree search techniques show promise, they are limited by their reliance on predefined heuristics and computationally expensive exploration strategies. We propose Policy-Guided Tree Search (PGTS), a framework that combines reinforcement learning with structured tree exploration to efficiently navigate reasoning paths. Our key innovation is a learned policy that dynamically decides between expanding, branching, backtracking, or terminating exploration, eliminating the need for manual heuristics or exhaustive search. Experiments across mathematical reasoning, logical deduction, and planning benchmarks demonstrate that PGTS achieves superior reasoning performance while significantly reducing computational costs compared to existing methods. These results establish PGTS as a scalable and effective solution for tackling complex reasoning tasks with LLMs.

## 1. Introduction

Large language models (LLMs) have demonstrated remarkable capabilities across diverse tasks, driven by advances in model architecture, parameter scaling, and pretraining data (Achiam et al., 2023; Team et al., 2023; Jiang et al., 2023; Dubey et al., 2024). However, these models consistently struggle with tasks requiring complex reasoning and planning (Valmeekam et al., 2022; Kambhampati, 2024; Kambhampati et al., 2024). In mathematical problem-solving, for instance, while LLMs excel at direct arithmetic calculations, they frequently falter when faced with multi-step word problems that demand strategic decomposition and careful planning (Kao et al., 2024; Huang & Chang, 2022).

These limitations extend to logical reasoning and real-world planning scenarios, where success depends on systematically breaking down complex problems into interconnected, actionable steps (Kambhampati, 2024).

Recent approaches to enhance LLM reasoning fall into three main categories. First, advanced prompting techniques like Chain-of-Thought (CoT) (Wei et al., 2022) and Least-to-Most prompting (Zhou et al., 2022) encourage step-by-step reasoning by generating intermediate steps. Second, verification-based methods aim to improve reasoning quality through step validation (Cobbe et al., 2021; Lightman et al., 2023; Li et al., 2022; Wang et al., 2024b; Zhang et al., 2024) or iterative refinement (Qu et al., 2024), using either self-evaluation (Qu et al., 2024) or external correctors (Havrilla et al., 2024). Third, tree-based search methods reframe reasoning as a planning problem, using reward signals to guide the exploration of reasoning paths (Feng et al., 2023; Yao et al., 2024; Besta et al., 2024; Hao et al., 2023; Xie et al., 2024b; Khalifa et al., 2023; Wang et al., 2024a).

While these approaches show promise, they face several key limitations. Heuristic search methods rely heavily on predefined rules and reward definitions, requiring significant expert knowledge. Additionally, their trial-and-error exploration process can be computationally expensive due to the vast space of possible reasoning steps. The challenge becomes even more pronounced when using self-evaluation as a reward signal, making the process even more resource-intensive (Feng et al., 2023; Hao et al., 2023; Xie et al., 2024b). Moreover, the ability of LLMs to effectively critique their own outputs and refine their responses remains an area of active research. This limitation is particularly pronounced in tasks requiring intricate planning and reasoning (Stechly et al., 2024; Huang et al., 2023; Hong et al., 2023). Furthermore, even when the exploration process identifies improved reasoning chains, distinguishing successful reasoning paths from failed ones without external guidance continues to pose a significant challenge (Qi et al., 2024).

To address these challenges, we propose Policy-Guided Tree Search (PGTS), a framework that integrates reinforcement learning with structured tree exploration. The core of PGTS is a learned policy that dynamically guides the reasoning process through four key actions: expanding current nodes, branching to alternative paths, backtracking to previous

---

[1]Independent Researcher. Correspondence to: Yang Li <yli.ml.research@gmail.com>.

*Proceedings of the $42^{nd}$ International Conference on Machine Learning*, Vancouver, Canada. PMLR 267, 2025. Copyright 2025 by the author(s).

states, or terminating exploration. This structured approach enables efficient navigation of reasoning paths, focusing computational resources on the most promising paths while avoiding unnecessary exploration of less relevant areas.

PGTS offers several key advantages over existing methods. First, its learned policy adapts dynamically to different tasks without requiring predefined heuristics. Importantly, training the policy does not require ground-truth reasoning chain annotations, making it more flexible and scalable. Second, the framework's backtracking capability allows recovery from suboptimal paths, addressing a common limitation of traditional search methods, such as Depth-Frist Search (DFS) and Breadth-First Search (BFS) (Yao et al., 2024; Xie et al., 2024b). Third, the terminate action prevents unnecessary exploration by halting the process once sufficient reasoning evidence has been gathered, significantly reducing computational overhead and addressing the "overthinking" phenomenon often observed in o1-like reasoning chains (Chen et al., 2024c). Finally, the integration of reinforcement learning with tree search creates an effective balance between exploiting known high-reward paths and exploring new alternatives. Collectively, these features make the PGTS framework a robust, efficient, and versatile solution for tasks requiring complex reasoning and planning.

Extensive experiments across mathematical reasoning, logical deduction, and planning benchmarks demonstrate the effectiveness of PGTS. Using LLaMA3.1-8B, PGTS achieves a 41.00% accuracy on MATH, significantly improving upon CoT's 34.40% while using only one-third of the tokens required by MCTS. These results establish PGTS as a practical and scalable solution for enhancing LLM reasoning capabilities across diverse problem domains.

## 2. Method

### 2.1. Problem Formulation

Language model-based reasoning can be formalized as a sequence generation problem with intermediate steps. Large language models, parameterized by $\theta$ and denoted as $p_\theta$, generate text autoregressively. Given an input prompt $x = [x_1, \ldots, x_n]$ consisting of tokens from a predefined vocabulary, the model produces a response $y = [y_1, \ldots, y_m]$. Each token $y_i$ is produced sequentially, conditioned on the prompt $x$ and all previously generated tokens $y_{<i}$. The probability of generating sequence $y$ can be expressed as:

$$p_\theta(y \mid x) = \prod_{i=1}^{m} p_\theta(y_i \mid x, y_{<i}). \qquad (1)$$

For reasoning tasks, $y$ typically comprises both intermediate reasoning steps and the final answer.

This autoregressive procedure naturally maps to a Markov Decision Process (MDP), defined as $(\mathcal{S}, \mathcal{A}, T, R, \gamma)$, where

the state $s \in \mathcal{S}$ represents the current context, including the prompt $x$ and generated tokens so far; the action $a \in \mathcal{A}$ corresponds the next token to be generated; the transition $s' = T(s, a)$ is deterministically defined by appending action $a$ to state $s$; the reward $R(s, a)$ evaluates the quality of each state-action pair; $\gamma$ denotes the discount factor, weighting immediate rewards over future ones. The objective of reasoning is to find an optimal sequence of actions $a^* = [a_1, \ldots, a_H]$ that maximizes the cumulative discounted rewards, $r = \sum_{h=1}^{H} \gamma^h R(s_h, a_h)$, where $H$ is the trajectory horizon. In practice, actions can represent tokens, phrases, or complete sentences to improve optimization efficiency (Feng et al., 2023; Zhao et al., 2024; Xie et al., 2024a; Yao et al., 2024; Wang et al., 2024a). We refer to this MDP formulation as LM-MDP.

### 2.2. Background: Tree Search for LLM Reasoning

Tree search methods offer a structured approach to explore and optimize reasoning paths in LLMs by systematically evaluating different sequences of reasoning steps. The reasoning process can be formalized as a tree $(\mathcal{V}, \mathcal{E})$, where each node $v \in \mathcal{V}$ corresponds to a state $s \in \mathcal{S}$, and each edge $(v_i, v_j) \in \mathcal{E}$ represents an action $a \in \mathcal{A}$. The root node represents the initial prompt $x$, while a path from the root to a leaf node denotes a complete reasoning chain $y$. To evaluate the quality of each reasoning step, a process reward model (PRM) $R(s, a)$ is employed, implemented either as a pretrained reward model (Lightman et al., 2023; Uesato et al., 2022; Wang et al., 2024b) or an LLM-based evaluator (Yao et al., 2024; Gao et al., 2024; Hao et al., 2023).

Various tree search algorithms have been developed to navigate these reasoning trees effectively. Depth-First Search (DFS) (Yao et al., 2024) and Breadth-First Search (BFS) (Xie et al., 2024b) offer systematic exploration strategies, while A* Search (Wang et al., 2024a) and Monte Carlo Tree Search (MCTS) (Feng et al., 2023; Zhao et al., 2024; Xie et al., 2024a) incorporate heuristics and sampling to balance exploration with exploitation. While these methods have proven effective in black-box optimization (Hart et al., 1968; Zhai & Gao, 2022; Malherbe et al., 2022; Wang et al., 2024c; 2020) and reinforcement learning (Silver et al., 2017; Kartal et al., 2019; Grill et al., 2020) contexts, their application to LLM reasoning introduces distinct challenges: First, the potential sequences of tokens, phrases, or sentences at each step create an enormous action space, making exhaustive exploration computationally infeasible. Second, feedback signals $R(s, a)$ are often sparse, noisy, or difficult to compute, particularly in the absence of ground-truth annotations for reasoning chains, complicating the evaluation of reasoning paths. Third, querying the LLM at each reasoning step incurs significant resource demands, amplifying the computational overhead of traditional tree search methods.

These challenges highlight the need for more efficient and

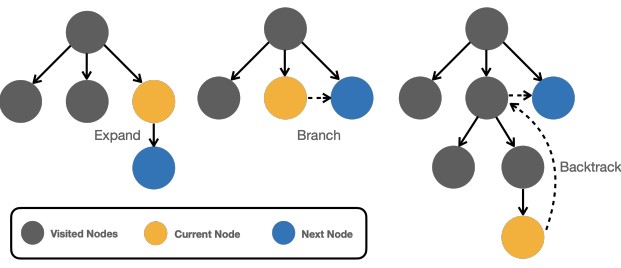

*Figure 1.* Expand, Branch and Backtrack actions in PGTS policy.

adaptive search strategies in LLM reasoning. To address these limitations, we propose the PGTS framework, which integrates a learned policy to dynamically guide the tree search process, balancing exploration and exploitation while reducing computational overhead.

## 2.3. Policy Guided Tree Search

### 2.3.1. OVERVIEW

Building on the tree search formulation for LLM reasoning, PGTS introduces a structured action space designed to navigate the reasoning tree effectively. Rather than relying on predefined search strategies, PGTS learns to dynamically select between four fundamental operations:

- **Expand**: Generate the next intermediate step in the reasoning chain by expanding the current node. This allows the policy to progress along promising reasoning paths.

- **Branch**: Explore alternative reasoning paths by branching to a sibling node from the current node. This enables the policy to consider alternative solutions when the current one is suboptimal.

- **Backtrack**: Revisit a previous node to explore alternative paths, enabling recovery from suboptimal reasoning trajectories. The backtrack operation involves multiple actions, as the policy specifies how many steps to backtrack.

- **Terminate**: Conclude the search process once sufficient evidence or a satisfactory reasoning chain is obtained, preventing unnecessary exploration.

These actions (denoted as $\mathbb{A}$) empower PGTS to adaptively explore and refine reasoning chains, striking a balance between exploring promising steps and exploiting valuable chains. This dynamic approach allows for efficient and targeted navigation of the reasoning tree, even in the presence of a vast reasoning steps. Please see Fig. 1 for an illustration.

In practice, we impose practical constraints to ensure computational feasibility. We limit the maximum depth of the reasoning tree, which naturally bounds the number of possible backtracking steps. To prevent unbounded exploration from branching actions, we also restrict the tree's breadth by capping the number of child nodes per parent. These constraints maintain a finite action space while preserving the

policy's ability to explore diverse reasoning paths. Please see Sec. 2.3.3 for detailed discussion about the implication of these depth and breadth limits.

The PGTS policy performs an exploratory walk on the underlying reasoning tree, which is initially unobserved and gradually revealed through exploration. This process can be modeled as a MDP operating within an environment represented by the complete reasoning tree. The state $\mathbb{S}$ corresponds to the portion of the tree revealed up to the current step, transitioning deterministically based on the chosen action $\mathtt{a} \in \mathbb{A}$. Note that this formulation differs from the previously introduced LM-MDP, as it operates at the level of tree exploration rather than token generation. Please see Sec. 2.3.2 for the formal definition of this Tree Search MDP.

Since the state $\mathtt{s} \in \mathbb{S}$ is naturally tree-structured, we implement the PGTS policy $\pi_\phi(\cdot \mid \mathtt{s})$ as a graph neural network. Specifically, we utilize the Graph Transformer architecture, GPS, proposed in (Rampášek et al., 2022). The GPS architecture utilizes both local message passing and global attention to capture both the local structural relationships and global context within the tree. Please see Sec. 2.3.4 for details about the PGTS policy.

To guide the reasoning process effectively, the reward design for PGTS balances two objectives: promoting high-quality reasoning chains while encouraging efficient exploration. We define immediate rewards based on task-specific metrics that evaluate the relevance and correctness of generated reasoning steps, $R(s, a)$, while introducing costs for exploration actions. This reward structure ensures the policy learns to navigate the reasoning tree efficiently, focusing computational resources on the most promising paths. Please see Sec. 2.3.2 for details about the reward design.

### 2.3.2. TREE SEARCH MDP

The tree search problem in PGTS can be formulated as a MDP, which we refer to as Tree Search MDP (TS-MDP), defined by the tuple $(\mathbb{S}, \mathbb{A}, \mathbb{T}, \mathbb{R}, \gamma)$. The MDP interacts with the underlying reasoning tree as its environment, with each reasoning problem corresponding to a new environment. Similar to the typical MDP settings, the environment is not directly observable to the policy, while it gradually reveals a part of the tree as the policy explores the environment.

The state space $\mathbb{S}$ represents the set of possible states in the tree exploration process, where a state $\mathtt{s} \in \mathbb{S}$ corresponds to the portion of the reasoning tree that has been revealed through previous interactions. The state includes the structure and content of all visited nodes, as well as the current position of the policy in the tree. Each node in the tree represents a partial reasoning path, which is equivalent to a state $s \in \mathcal{S}$ in the LM-MDP. Node features are derived from the hidden states extracted from the target LLM, specifically the hidden state from the final layer corresponding to the

last generated token. Each edge in the tree captures the parent-child relationship established by the reasoning step $a \in \mathcal{A}$. The edge features include the immediate reward $R(s, a)$, reflecting the quality of the reasoning step $a$. Together, the structure of nodes and edges forms a dynamic representation of the reasoning tree, allowing the PGTS policy to effectively navigate and optimize the exploration.

The action space $\mathbb{A}$ comprises the structured actions available to the PGTS policy: expand, branch, backtrack, and terminate. Each action transitions the policy to a new state by modifying the observed reasoning tree. For a reasoning tree with a maximum depth of $D$, the backtrack operation includes $D - 1$ distinct actions, corresponding to backtracking between 1 and $D - 1$ steps. Notably, backtracking to the root node (corresponding to the prompt $x$) is invalid, as the root node lacks sibling nodes for exploration.

The transition function $\mathbb{T}$ determines how the state transitions from $s$ to $s'$ given an action $a \in \mathbb{A}$. The transitions in the TS-MDP are deterministic, as applying an action updates the tree structure in a predefined manner. The reasoning process starts from a single root node, representing the initial state, and terminates when the policy selects the terminate action or when the exploration budget, i.e., the maximum number of tree search steps, is exhausted. As the policy updates the state, it records the node features and edge features, and update the current node accordingly to reflect the latest position of the policy in the reasoning tree.

The reward function $\mathbb{R}(s, a)$ assigns a scalar reward for executing action $a$ in state $s$. For each action type, the reward is defined as follows:

- **Expand**: Let $s_d \in \mathcal{S}$ be the partial reasoning path at the current node and $a_d \in \mathcal{A}$ be the new reasoning step generated by the expand action, where $d$ denotes the depth of the current node. The reward is defined as

$$\mathbb{R}(s, a) = R(s_d, a_d) - \mathbb{C}(a), \quad (2)$$

where $R(s_d, a_d)$ evaluates the quality of the new reasoning step, and $\mathbb{C}(a)$ denotes the cost of the expand action.

- **Branch**: When the policy selects the branch action to explore an alternative sibling node, let $s_d$ and $s'_d$ denote the partial reasoning paths corresponding to the current node and its sibling node respectively, with shared parent $s_{d-1}$, the reward is defined as

$$\mathbb{R}(s, a) = R(s'_{d-1}, a'_{d-1}) - R(s_{d-1}, a_{d-1}) - \mathbb{C}(a). \quad (3)$$

Here, $R(s'_{d-1}, a'_{d-1}) - R(s_{d-1}, a_{d-1})$ quantifies the potential improvement in reasoning by exploring the alternative path, while $\mathbb{C}(a)$ reflects the cost of branching.

- **Backtrack**: For the backtrack action, which reverts to a previously visited node, all intermediate rewards along the backtracked path are revoked. Suppose the backtrack

action reverts $K$ steps. Denote the partial reasoning paths for the current node and previous node as $s_d$ and $s_{d-K}$, respectively. The reward $\mathbb{R}(s, a)$ is defined as

$$R(s_{d-K-1}, a'_{d-K-1}) - \sum_{k=1}^{K} R(s_{d-k}, a_{d-k}) - \mathbb{C}(a), \quad (4)$$

where $\sum_{k=1}^{K} R(s_{d-k}, a_{d-k})$ represents the accumulated reward along the backtracked path, $R(s_{d-K-1}, a'_{d-K-1})$ captures the reward from the new path, and $\mathbb{C}(a)$ indicates the cost of the backtrack action.

- **Terminate**: When the policy selects the terminate action to end the reasoning process, let $s_d$ denote the final reasoning path at the terminal node, the reward is

$$\mathbb{R}(s, a) = R(s_d) - \mathbb{C}(a), \quad (5)$$

where $R(s_d)$ measures the overall quality or correctness of the reasoning chain at the terminal node, and $\mathbb{C}(a)$ represents the cost of concluding the search. The reward $R(s_d)$ can be derived from the commonly used outcome reward model (ORM), and we can even compare the final answer to the ground truth, as the reward is only used during the training of the PGTS policy.

The cost function $\mathbb{C}(a)$ plays a pivotal role in guiding the PGTS policy by assigning penalties to actions based on their computational or logical expense. These costs act as a mechanism to balance efficiency and reasoning quality. Actions such as expand and branch typically incur lower costs, as they change the reasoning tree locally. In contrast, the backtrack actions involve higher cost, discouraging excessive reversions unless the potential improvements justify the expense. The terminate action, which concludes the reasoning process and avoids further resource usage, generally carries negligible or zero cost. In our implementation, $\mathbb{C}(a)$ is treated as a hyperparameter that can be tuned to align the system's behavior with task-specific requirements or computational constraints. By appropriately setting these costs, the system fosters a balance between computational efficiency and reasoning complexity. This cost structure works in tandem with the reward function, which prioritizes accurate, coherent, and relevant reasoning paths while penalizing unproductive exploration. Together, the rewards and costs enable the PGTS policy to navigate reasoning trees effectively, balancing exploration and exploitation, and adapting flexibly to diverse tasks and constraints.

### 2.3.3. CONSTRAINED ACTION SAMPLING

The reasoning tree enforces explicit depth and breadth limits to maintain computational feasibility, which naturally constrains the validity of actions in different states. For expand actions, validity is determined by the depth limit and answer state: the expand action becomes invalid when the

current node has reached the maximum depth limit or when a final answer has already been generated at the current node. Branch and backtrack actions face similar structural constraints. These actions are prohibited at the root node as it has no parent or siblings. They are also invalid when the parent node has reached its maximum allowed children, preventing unbounded tree growth. For backtrack specifically, a node at depth $d$ can only backtrack up to $d - 1$ steps, ensuring the action remains within the explored portion of the tree. The terminate action maintains the simplest constraint, becoming valid only when the current node has reached a final answer or the maximum depth limit has been reached.

These structural constraints ensure the reasoning process remains well-defined and computationally tractable while guiding the policy toward meaningful exploration paths. To implement these constraints efficiently, we represent them as a $D + 2$ dimensional binary vector, where $D$ indicates the depth limit of the reasoning tree. Each dimension corresponds to the validity of a specific action: the first dimension for expand, the second for branch, the next $D - 1$ dimensions for different backtrack depths, and the final dimension for terminate. This constraint vector serves dual purposes: it augments the policy's input state to inform action selection, and it masks invalid actions from the policy's output distribution, ensuring only valid actions can be sampled. Through this mechanism, the policy naturally learns to navigate within the feasible action space while maintaining the structural integrity of the reasoning tree.

### 2.3.4. POLICY DESIGN FOR PGTS

The effectiveness of PGTS heavily depends on its ability to process and navigate reasoning tree. In our TS-MDP formulation, the state is naturally represented as a tree where each node corresponds to a partial reasoning path. To effectively learn from this structured data, we need a policy architecture that can capture both local relationships between reasoning steps and global patterns across the entire tree.

We design our policy using a graph-based architecture that processes two types of features. Node features are derived from the hidden states of the target LLM, capturing the semantic content of each reasoning step. Edge features encode the intermediate rewards of reasoning transitions, quantifying the quality and relationships between consecutive steps. This rich feature representation enables the policy to assess both the local quality of individual steps and their contribution to the overall reasoning path.

To process these features effectively, we employ a Graph Transformer-based model using the GPS framework (Rampášek et al., 2022). This architecture combines local message passing operations from graph neural networks with global attention mechanisms from transformers. The local operations capture step-by-step reasoning relationships, while global attention helps maintain coherence across the entire reasoning process. We further enhance the structural understanding by incorporating random-walking structure embeddings (Dwivedi et al., 2021), which encode each node's position and connectivity within the tree.

The policy network processes these inputs through a sequence of GPS layers, each layer aggregating both local and global information. The final representation of the current node is concatenated with the action constraints vector to ensure awareness of valid actions. This combined representation then passes through linear layers to produce logits for a categorical distribution over the $D + 2$ possible actions. Invalid actions are masked out during sampling, ensuring the policy's decisions respect the tree's structural constraints.

We implement the value network using the same architectural backbone, sharing the GPS layers with the policy network to maintain consistent representation learning. The value network differs only in its final layers, which produce a scalar estimate of the expected cumulative reward. This shared structure allows both networks to leverage the same learned representations of the reasoning tree while serving their distinct purposes in the decision-making process.

### 2.3.5. TRAINING

The training process for the PGTS policy aims to enhance reasoning effectiveness while minimizing unnecessary exploration. This is accomplished by optimizing the policy through reinforcement learning, with rewards designed to promote both accuracy and efficiency. The training iteratively refines the policy's ability to navigate reasoning trees and select actions that yield high-quality solutions.

We adopt Proximal Policy Optimization (PPO) (Schulman et al., 2017) as our training algorithm due to its stability and sample efficiency. Starting from randomly initialized weights, the policy interacts with reasoning tasks in episodes, learning to select actions that maximize cumulative rewards. When sampling actions during training, we apply the constraint mask to zero out logits of invalid actions, ensuring the decisions respect the tree's structural limits. The remaining logits are normalized to form a valid categorical distribution over permitted actions. This mechanism allows the policy to learn feasible exploration strategies while maintaining the integrity of the reasoning tree.

To encourage efficient exploration, we incorporate entropy regularization into the policy loss. This ensures that the policy maintains a balance between exploiting known high-reward paths and exploring less certain but potentially rewarding alternatives. Additionally, the cost components $\mathbb{C}(\mathtt{a})$ in the reward function are tuned to discourage excessive backtracking or branching, guiding the policy toward concise, meaningful reasoning paths. Please refer to Algorithm 1 for the complete training procedure.

## 3. Related Works

**LLM Reasoning**   LLM reasoning has advanced through techniques such as CoT (Wei et al., 2022), ToT (Yao et al., 2024), and programmatic reasoning paradigms (Chen et al., 2022; Sel et al., 2023), fostering structured and iterative problem-solving. Recent innovations include heuristic search methods like MCTS (Feng et al., 2023; Hao et al., 2023) and A* search (Wang et al., 2024a). Building on these developments, our PGTS framework integrates learned policies to improve search efficiency and reasoning performance. For a detailed review of related approaches and their connection to inference-time scaling, please refer to Sec. B.

**Graph Transformers**   Graph Transformers (GTs) have emerged as powerful architectures for processing graph-structured data, building upon the success of Transformers in other domains. These models are particularly attractive due to their ability to address fundamental limitations of traditional Message Passing Neural Networks (MPNNs), such as over-smoothing and over-squashing issues (Alon & Ya-hav, 2020; Topping et al., 2021). Various GT architectures have been proposed, from the initial Fully-connected Graph Transformer with basic positional encodings (Dwivedi & Bresson, 2020), to more sophisticated designs like SAN with invariant eigenvector aggregation (Kreuzer et al., 2021), and Graphormer with distance-based encodings (Ying et al., 2021). GraphTrans (Wu et al., 2021) introduces the first hybrid architecture, which combines local message passing with global attention mechanisms. GPS (Rampášek et al., 2022) systematically investigates and integrates different components of GTs, offering a modular and scalable framework. In this work, we implement the PGTS policy using GPS layers given its ability to effectively combine local and global information while maintaining linear complexity.

## 4. Experiments

In this section, we showcase the flexibility and effectiveness of our PGTS framework across diverse problem domains, including mathematical reasoning, commonsense reasoning, logical reasoning, and real-world planning. For mathematical reasoning, we evaluate our framework on the GSM8K (Cobbe et al., 2021), MATH500 (Hendrycks et al., 2021; Lightman et al., 2023), and AQUA (Ling et al., 2017) datasets, using 4-shot settings for GSM8K and MATH500, and a 10-shot setting for AQUA. The in-context learning (ICL) examples are adapted from OpenCompass (Contributors, 2023) for GSM8K and MATH500, and from LLM-Reasoner (Hao et al., 2024) for AQUA. For commonsense reasoning, we evaluate StrategyQA (Geva et al., 2021) in a 5-shot setting, with ICL examples also adapted from Open-Compass. For logical reasoning, we evaluate the PrOntoQA (Saparov & He, 2022) dataset for logical deduction in a 5-shot setting and the GPQA (Rein et al., 2023) dataset for graduate-level multiple-choice questions in a 0-shot setting. Finally, for the planning task, we evaluate our framework on the Blocksworld benchmark (Valmeekam et al., 2022).

Across all datasets, we define a single reasoning step as one sentence, maintaining consistency and simplicity. Unlike RAP (Hao et al., 2023), which incorporates a world model to simulate environment states after each action, our approach directly focuses on reasoning in the generated text without state modeling. For detailed dataset descriptions and reasoning setups, refer to Sec. C.

We compare PGTS against CoT and MCTS baselines. MCTS inherently explores multiple reasoning chains by traversing different paths during search, while we enhance CoT and our PGTS with self-consistency (SC) (Wang et al., 2022). Specifically, CoT aggregate outcomes from multiple chains using majority voting, whereas MCTS and PGTS utilize weighted voting based on the reward of each reasoning chain. Additionally, we report MCTS results for the highest-reward trajectory. Since one of our primary goals is to reduce reasoning costs, we exclude MCTS approaches that rely on self-evaluation as intermediate rewards (Xie et al., 2024a; Gao et al., 2024), as they are computationally expensive. Instead, we simply use the likelihood of each reasoning step as the intermediate reward, $R(s, a)$. An oracle setting is included for MCTS, allowing it to access task rewards by comparing generated answers with ground truths during search. For both MCTS and PGTS approaches, the tree breadth is limited to 4 child node per parent. For detailed descriptions of the baselines, refer to Sec.D, and for ablations on the reasoning tree constraints, see Sec.4.1.

For PGTS, we train the policy using up to 1,000 examples from the training split of each dataset. This highlights the sample efficiency of our approach, as 1,000 examples suffice to learn an effective policy. For ablations on training settings, refer to Sec. 4.1. The policy architecture consists of two GPS layers followed by a single linear layer for action and value prediction. To simplify experiments, the action cost $\mathbb{C}(\texttt{a})$ is fixed across datasets, with values of $0.1$, $0.2$, $0.5$, and $0.0$ for expand, branch, backtrack, and terminate actions, respectively. While dataset-specific tuning could further enhance performance, we leave this for future exploration. See Sec. E for training details of our PGTS policy.

Table 1 presents evaluation results for the LLaMA3.1-8B and LLaMA3.1-70B models, both configured to generate reasoning steps with a temperature of $0.6$ and top_p of $0.9$. Across all datasets, PGTS consistently outperforms CoT, demonstrating the effectiveness of its learned policy in guiding reasoning steps. The inclusion of SC further boosts PGTS performance. For instance, on the MATH dataset, PGTS improves accuracy from CoT's 34.40% to 41.40% in the 8B setting, illustrating its ability to explore high-quality reasoning chains. Similarly, on Blocksworld, PGTS-SC

*Table 1.* Evaluation results of LLaMA3.1-8B and LLaMA3.1-70B on various datasets across multiple reasoning tasks: Mathematical reasoning (GSM8K, MATH, AQUA), Commonsense reasoning (StrategyQA), Logical reasoning (PrOntoQA, GPQA), and Planning tasks (Blocksworld with 4 and 8 steps). SC4 and SC8 denote self-consistency voting over 4 and 8 sampled chains, respectively. MCTS (Best) reports results for the reasoning chain with the highest reward, while MCTS (Agg) presents results aggregated over multiple reasoning chains using weighted voting on the final answer. MCTS (Oracle) compares final answer with groundtruth as an additional reward.

|  |  | Mathematical | | | Com. Sense | Logical | | Planning | |
| --- | --- | --- | --- | --- | --- | --- | --- | --- | --- |
|  |  | GSM8K | MATH | AQUA | StrategyQA | PrOntoQA | GPQA | BW (4) | BW (8) |
| LLaMA3.1-8B | CoT | 83.47 | 34.40 | 51.57 | 74.20 | 69.40 | 14.65 | 22.37 | 2.10 |
|  | CoT (SC4) | 87.79 | 42.20 | 53.94 | 77.10 | 73.00 | 15.66 | 22.37 | 2.79 |
|  | CoT (SC8) | 89.84 | 48.20 | 55.12 | 77.20 | 74.60 | 15.15 | 26.32 | 2.79 |
|  | MCTS (Best) | 86.13 | 43.80 | 60.63 | 79.00 | 74.20 | 34.34 | 28.95 | 6.29 |
|  | MCTS (Agg) | 87.72 | 46.00 | 59.45 | 79.50 | 74.20 | 32.83 | 28.95 | 6.29 |
|  | MCTS (Oracle) | 88.78 | 51.40 | 64.96 | 84.40 | 74.80 | **34.85** | 34.21 | 6.29 |
|  | PGTS | 85.29 | 41.00 | 60.63 | 77.70 | 68.20 | 18.69 | 27.63 | 3.50 |
|  | PGTS (SC4) | 89.61 | 47.00 | **66.93** | 83.80 | 74.40 | 22.73 | 35.53 | 4.90 |
|  | PGTS (SC8) | **89.99** | **52.20** | 66.54 | **85.70** | **77.40** | 27.78 | **36.84** | **6.99** |
| LLaMA3.1-70B | CoT | 91.66 | 53.80 | 72.83 | 83.60 | 92.00 | 20.20 | 39.47 | 18.88 |
|  | CoT (SC4) | 92.49 | 58.60 | 72.44 | 85.10 | 93.60 | 19.78 | 50.00 | 20.28 |
|  | MCTS (Best) | 91.28 | 56.20 | 76.38 | 85.70 | 95.20 | 41.92 | 50.00 | 22.38 |
|  | MCTS (Agg) | 92.27 | 57.20 | 77.17 | 86.70 | 95.20 | 43.94 | 50.00 | 22.37 |
|  | MCTS (Oracle) | **93.10** | **66.00** | **82.68** | 90.70 | 96.00 | **47.47** | **61.80** | 24.47 |
|  | PGTS | 91.05 | 54.77 | 73.23 | 85.50 | 91.40 | 32.18 | 46.05 | 22.38 |
|  | PGTS (SC4) | 92.54 | 59.65 | 79.92 | **91.00** | **96.80** | 36.36 | 50.00 | **25.87** |

achieves an accuracy of 36.84%, compared to CoT-SC's 26.32%, showcasing its superior handling of multi-step reasoning tasks. While both CoT and PGTS benefit from SC, PGTS exhibits more pronounced improvements, underscoring its ability to generate diverse, high-quality chains suitable for aggregation. Please see Sec. G for examples of the generated reasoning chains.

When compared to MCTS, PGTS demonstrates competitive performance across most tasks. For example, PGTS without SC achieves comparable accuracy to MCTS on GSM8K, MATH, and StrategyQA and surpasses MCTS when augmented with SC. However, one exception is the GPQA task, where PGTS underperforms MCTS. This can be attributed to the inherent complexity of GPQA, which includes diverse topics and is curated to challenge even non-expert humans. Moreover, the limited training data available for PGTS, given the task complexity, makes it difficult to fully capture the intricacies of GPQA.

A key advantage of PGTS over MCTS is its computational efficiency. As shown in Figure 2, MCTS incurs significantly higher token costs due to its exhaustive search over reasoning chains. For instance, on MATH, MCTS requires 16.25 times more tokens than CoT, whereas PGTS achieves competitive performance with only a 5.28 times increase. Similarly, on GSM8K, MCTS uses 13.33 times the tokens of CoT, while PGTS requires just 1.29 times. These reductions in token usage make PGTS more practical for real-world applications where computational costs are critical. Although

SC requires generating additional tokens for aggregating outcomes over multiple reasoning chains, these operations are highly parallelizable, ensuring that PGTS with SC remains computationally feasible despite the additional generations. Notably, the overhead introduced by the PGTS policy is negligible compared to the cost of LLM generation, thanks to its lightweight architecture.

### 4.1. Ablations

**Training Examples** In the main results, we train the PGTS policy using up to 1,000 examples. Here, we demonstrate the convergence behavior of the training process. Figure 3 presents the training curve with evaluation results at intermediate checkpoints, showing that the policy converges quickly and plateaus at approximately 1,000 examples.

**Tree Constraints** In our main results, we limit the tree breadth to 4 child nodes per parent. Table 2 shows evaluation results on AQUA with varying tree breadths. Both MCTS and PGTS achieve better performance with broader trees, as expected, since a broader tree facilitates more diverse reasoning chains. However, broader trees also generate more tokens, so we use a breadth of 4 in our main results to balance accuracy and reasoning cost. For tree depth, we set an upper bound that ensures all examples can reach the terminal state (i.e., the final answer), with the terminate action allowing early stopping as needed.

**Policy Network** We implement our PGTS policy using GPS layers, which combine local message passing and

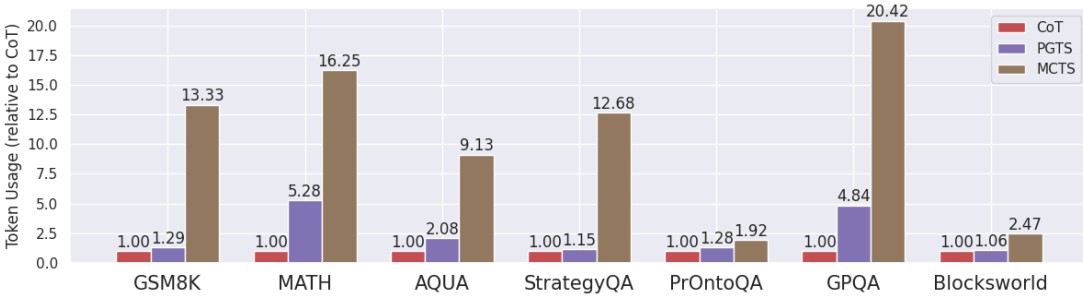

*Figure 2.* Comparison of generated token counts for LLaMA3.1-8B, normalized relative to the CoT method.

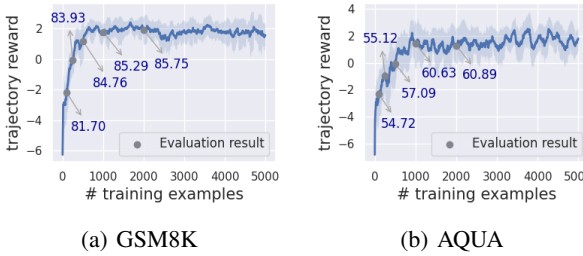

(a) GSM8K          (b) AQUA

*Figure 3.* Trajectory reward along training, with evaluation results at intermediate checkpoints.

*Table 2.* AQUA results with different tree breadth.

| | | 2 | 4 | 8 |
|---|---|---|---|---|
| Evaluation Results | PGTS | 55.91 | 60.63 | 61.47 |
| | MCTS | 57.08 | 59.45 | 62.60 |
| Generated Tokens | PGTS | 252.16 | 283.24 | 370.54 |
| | MCTS | 399.13 | 1241.37 | 3282.41 |

*Table 3.* Ablation studies on policy implementation.

| | AQUA | GSM8K |
|---|---|---|
| GPS | **60.63** | **85.29** |
| GPS (w/o edge features) | 50.79 | 81.96 |
| GPS (w/o global attention) | 54.72 | 84.15 |
| GPS (w/o local MPNN) | 58.76 | 84.78 |
| SAN | 60.24 | 82.56 |
| SLM | 53.94 | 82.87 |
| LLM Agent | 55.51 | 83.32 |

advanced graph modeling techniques could further enhance results. SLM and LLM Agent perform worse, underscoring the strengths of graph-based approaches over language-only methods. We hypothesize that techniques such as improved prompt engineering or memory-augmented agents (Chen et al., 2024b) could boost agent-based performance, which we leave for future work.

## 5. Conclusion

In this work, we introduced Policy-Guided Tree Search (PGTS), a novel framework for reasoning with large language models that combines the efficiency of policy-guided exploration with the structured advantages of tree search. PGTS dynamically allocates inference resources, prioritizing promising reasoning paths for targeted exploration, leading to significant improvements in inference efficiency and the ability to tackle complex reasoning tasks. Moreover, PGTS addresses the "overthinking" problem commonly observed in many o1-like models (Chen et al., 2024c), where excessive reasoning steps are generated for simple problems.

A key paradigm shift in PGTS is treating the LLM as an environment rather than a policy, enabling external decision-making components to guide reasoning processes more effectively. While the current implementation uses simple log-likelihood-based rewards as intermediate feedback, future work could incorporate more sophisticated reward mechanisms, such as LLM self-evaluation or task-specific metrics, to provide richer guidance during the search and further enhance reasoning performance. PGTS represents a significant step toward more efficient and structured inference-time reasoning with LLMs.

global attention to extract node features effectively. The reasoning tree also incorporates the immediate reward $R(s, a)$ as edge features to inform decision-making. Table 3 presents ablation studies evaluating the importance of each component in the GPS-based policy and comparing it to alternative policy implementations. **SAN** replaces GPS layers with a different graph transformer architecture proposed in (Kreuzer et al., 2021). **SLM** replaces the graph-based policy with a small language model (`distilbert-base-uncased`), which processes the previous reasoning trajectory to predict the next action. **LLM Agent** prompts the same target LLM to predict the exploration action, aligning with an agentic approach where the LLM assumes the role of an autonomous agent collaborating with others to solve the problem. Details of these policy implementations are provided in Sec. F. The ablation results demonstrate that the full GPS policy achieves the best performance, highlighting the effectiveness of integrating local message passing, global attention, and edge features. Removing edge features or global attention results in significant performance drops, especially on AQUA, emphasizing their importance for reasoning. While SAN performs competitively, it falls short of GPS, suggesting that

## Impact Statement

This work aims to advance the field of reasoning with large language models (LLMs) by introducing the Policy-Guided Tree Search (PGTS) framework. By improving inference efficiency and addressing challenges such as overthinking in reasoning processes, PGTS contributes to the broader objective of enhancing LLM usability in complex reasoning tasks.

The proposed framework has the potential to positively impact areas where efficient reasoning and decision-making are critical, including education, scientific discovery, and AI-driven assistance. However, it is essential to acknowledge certain ethical considerations and limitations. PGTS does not explicitly address the faithfulness of generated reasoning chains, meaning the reasoning paths may not align with human-understandable logic or the true underlying reasoning process. This limitation could inadvertently lead to misleading outputs in high-stakes applications.

Future work exploring richer reward signals, such as self-evaluation or task-specific metrics, could mitigate these concerns and ensure greater alignment with human expectations. While PGTS optimizes reasoning efficiency, care must also be taken to ensure it is applied responsibly, particularly in applications requiring transparency and trust in AI reasoning processes.

In summary, this work represents a meaningful step toward improving LLM reasoning capabilities, with both ethical implications and societal benefits that warrant further exploration and thoughtful application.

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

# A. Training Algorithm

---

**Algorithm 1** Policy Training in TS-MDP

---

**Require:** Reasoning tasks $\mathcal{T}$, depth limit $D$, breadth limit $B$, reward function $R(s, a)$, cost function $\mathbb{C}(a)$
**Ensure:** Trained policy $\pi(a|s)$ and value function $V(s)$
 1: **Initialize:** Parameters of policy $\pi_\phi$ and value function $V_\psi$
 2: **while** not converged **do**
 3:     Sample a task from $\mathcal{T}$, and initialize the root node $s_0$
 4:     **for** $t = 0, 1, \ldots, T_{\max}$ **do**
 5:         Extract valid actions and construct the constraints vector $c$
 6:         Compute policy logits $\pi \leftarrow \pi_\theta(s_t, c)$
 7:         Sample action $a_t \sim \text{Categorical}(\pi)$
 8:         Execute action $a_t$ and observe next state $s_{t+1}$, reward $r_t$, and constraints $c_{t+1}$
 9:         Store transition $(s_t, a_t, r_t, s_{t+1}, c_{t+1})$ in replay buffer $\mathcal{B}$
10:         **if** $a_t$ is Terminate **then**
11:             Break
12:         **end if**
13:     **end for**
14:     **Compute Returns and Advantages:**
15:     **for** each sampled trajectory in $\mathcal{B}$ **do**
16:         Compute discounted returns: $G_t = \sum_{j=0}^{\infty} \gamma^j r_{t+j}$
17:         Compute advantages: $A_t = G_t - V_\psi(s_t)$
18:     **end for**
19:     **Policy Update:**
20:     Compute policy loss $\mathcal{L}_\pi = -\mathbb{E}\big[\log \pi_\phi(a_t|s_t, c_t) \cdot A_t\big] - \omega \cdot H(\pi_\phi(a_t|s_t, c_t))$
21:     Update policy parameters: $\phi \leftarrow \phi - \eta_\pi \nabla_\phi \mathcal{L}_\pi$
22:     **Value Function Update:**
23:     Compute value loss $\mathcal{L}_V = \mathbb{E}\big[(V_\psi(s_t) - G_t)^2\big]$
24:     Update value parameters: $\psi \leftarrow \psi - \eta_V \nabla_\psi \mathcal{L}_V$
25: **end while**

---

# B. Related Works

**LLM Reasoning** In the context of LLM reasoning, Chain-of-Thought (CoT) resoning (Wei et al., 2022) serves as a foundational technique that decomposes problems into intermediate steps, facilitating step-by-step reasoning that mimics human problem-solving strategies. Since its introduction, numerous enhancements to CoT have been proposed, including Zero-Shot-CoT (Kojima et al., 2022), Self-Consistency with CoT (Wang et al., 2022), AutoCoT (Zhang et al., 2022), and VerifyCoT (Zhao et al., 2023), among others. Least-to-Most Prompting (Zhou et al., 2022) represents another influential paradigm that iteratively constructs solutions by dividing complex problems into simpler tasks. Building on this problem decomposition paradigm, Program-of-Thought (Chen et al., 2022), Chain-of-Code (Li et al., 2023), Buffer-of-Thought (Yang et al., 2024), and algorithm-of-Thought (Sel et al., 2023) incorporate programmatic reasoning steps to further enhance problem-solving. Tree-of-Thought (Yao et al., 2024) extends CoT by organizing reasoning processes into a tree structure, where each node represents a partial solution and edges signify transitions. However, ToT faces challenges with large search spaces for deep trees, leading to computational inefficiencies. To address this, Graph-of-Thought (Besta et al., 2024) generalizes ToT by modeling reasoning as a graph, allowing dynamic path selection, backtracking, and aggregation of information across multiple paths. Meanwhile, Skeleton-of-Thought (Ning et al., 2024) reduces latency by generating a skeleton outline of answers before completing the details in parallel. Recent research has also explored heuristic search techniques, such as Monte Carlo Tree Search (MCTS) (Feng et al., 2023; Hao et al., 2023) and A* Search (Wang et al., 2024a), to identify optimal reasoning chains. Building on these advancements, our proposed PGTS approach enhances heuristic search by integrating a learned policy to guide the search process, enabling more efficient exploration.

Beyond directly eliciting reasoning capabilities from LLMs, external components have been developed to verify reasoning steps (Cobbe et al., 2021; Lightman et al., 2023; Li et al., 2022; Wang et al., 2024b; Zhang et al., 2024) or refine generated reasoning through iterative revision (Qu et al., 2024; Havrilla et al., 2024). Additionally, iterative bootstrapping techniques

(Zelikman et al., 2022; 2024; Singh et al., 2023; Yuan et al., 2023) have been explored to progressively enhance LLM reasoning performance. These approaches are complementary to our method and can be integrated to achieve further improvements.

**Inference Time Scaling**    Optimizing inference-time compute has emerged as a critical area of research for improving the performance of LLMs. Unlike the traditional focus on scaling model parameters during training, inference-time scaling explores strategies to leverage additional compute during inference to enhance model outputs. One foundational approach in inference-time scaling is best-of-N sampling, where multiple completions are generated for a given prompt, and the best response is selected using a secondary mechanism, such as a reward model (Lightman et al., 2023). Recent studies have sought to formalize the relationship between inference compute and performance, proposing scaling laws for inference-time optimization (Snell et al., 2024; Wu et al., 2024; Chen et al., 2024a; Brown et al., 2024). These works show that carefully allocating inference resources, such as by balancing the number of samples with selection quality, can yield significant gains in model performance. Our proposed PGTS approach offers an efficient alternative to traditional inference-time scaling methods. Rather than passively sampling multiple outputs, the policy actively and dynamically decides when and where to allocate additional inference resources. This targeted approach enables more strategic exploration, focusing compute on the most promising reasoning paths.

# C. Evaluation Datasets

**GSM8K**    GSM8K (Cobbe et al., 2021) is a benchmark dataset designed to assess mathematical reasoning skills in language models. It comprises approximately 8,500 high-quality grade school math problems that require multi-step reasoning to arrive at the correct solution. We adhere to the original train-test split and evaluate models in a 4-shot setting, using in-context examples and a prompt template adapted from OpenCompass (Contributors, 2023), which provide representative coverage of various problem types.

During generation, each reasoning step is defined as a single sentence. To manage computational complexity, we set the tree breadth limit to 4, meaning each parent node can have up to 4 child nodes, and the depth limit to 16, which suffices to reach terminal nodes for most questions in this dataset. Additionally, the PGTS policy is configured to allow a maximum of 64 reasoning steps, balancing efficiency with the ability to explore complex reasoning chains.

**MATH500**    MATH500 (Hendrycks et al., 2021; Lightman et al., 2023) is a subset of the MATH dataset, designed to evaluate advanced mathematical problem-solving capabilities. It encompasses questions from high school and undergraduate-level topics, including calculus, geometry, linear algebra, and number theory. We use the original training split to train our PGTS policy and evaluate the framework in a 4-shot setting with in-context examples sourced from OpenCompass.

Each reasoning step is defined as a single sentence or a single line. To control computational complexity, we set the tree breadth limit to 4, the depth limit to 64, and the maximum reasoning steps for PGTS to 256, ensuring sufficient capacity for exploring complex mathematical reasoning chains.

**AQUA**    AQUA (Ling et al., 2017) is a dataset comprising algebraic word problems and reasoning-based multiple-choice questions. Its diverse question formats and logical reasoning challenges make it an effective benchmark for assessing models' general problem-solving abilities. We adhere to the original train-test split and evaluate our framework using a 10-shot setting, with in-context examples sourced from LLM-Reasoner (Hao et al., 2024), enabling the model to address problems involving numerical reasoning and reasoning by elimination.

Each reasoning step is defined as a single sentence. To manage computational complexity, we set the tree breadth limit to 4, the depth limit to 16, and the maximum reasoning steps for PGTS to 64, ensuring adequate exploration capacity for solving the dataset's challenges.

**StrategyQA**    StrategyQA (Geva et al., 2021) is a commonsense reasoning dataset designed to evaluate models' ability to answer binary (yes/no) questions by applying implicit multi-step reasoning strategies. The dataset features diverse and challenging questions where the intermediate reasoning steps are not explicitly provided. We randomly sample 1,000 examples for testing, with the remaining examples used to train the PGTS policy. Evaluation is conducted in a 5-shot setting, leveraging examples from OpenCompass to emphasize the model's capacity to infer and synthesize information across multiple reasoning steps.

Each reasoning step is defined as a single sentence. To balance computational efficiency and exploration, we set the tree breadth limit 4, the depth limit to 16, and the maximum reasoning steps for PGTS to 64, ensuring sufficient capacity to handle the dataset's complexity.

**PrOntoQA**    PrOntoQA (Saparov & He, 2022) is a benchmark designed to evaluate logical reasoning and deductive inference. It consists of structured questions that require precise logical deductions based on predefined rules and premises. We adopt the same dataset splits as used in RAP (Hao et al., 2023) and evaluate the model in a 5-shot setting to assess its ability to perform deductive reasoning while adhering to strict logical constraints. This dataset serves as a valuable tool for testing structured reasoning in a controlled environment.

Each reasoning step is defined as a single sentence. To balance computational efficiency and thorough exploration, we set the tree breadth limit to 4, the depth limit to 16, and the maximum reasoning steps for PGTS to 64, ensuring sufficient capacity to tackle the dataset's logical challenges.

**Blocksworld**    Blocksworld (Valmeekam et al., 2022) is a benchmark for evaluating real-world planning tasks, requiring models to reason over sequences of actions to achieve specific goals in a simulated block environment. The dataset assesses the ability to plan and execute multi-step strategies efficiently. We evaluate two versions that require 4 and 8 reasoning steps, using the same dataset splits and prompt template as RAP (Hao et al., 2023). Specifically, split-v1 is used for training and split-v2 for testing, with a 4-shot setup. In the easy setup, the in-context examples include the same number of reasoning steps as the test problems.

Unlike RAP, which employs a world model to predict the state after each action, we directly generate entire action trajectories without modeling intermediate states. Each reasoning step corresponds to one action described in a single sentence. We set the breadth limit to 4 and the depth limit to match the required number of reasoning steps for each version. The maximum reasoning steps for PGTS are capped at 32.

## D. Baseline Methods

**Chain-of-Thought**    CoT is a reasoning strategy that decomposes complex tasks into a sequence of intermediate steps, allowing models to reason step-by-step towards a final answer. To enhance the robustness of CoT, we incorporate self-consistency (SC) (Wang et al., 2022), where multiple reasoning chains are generated for the same input, and the outcomes are aggregated via majority voting. This voting scheme helps mitigate errors by leveraging diverse reasoning paths.

**Monte Carlo Tree Search**    MCTS has been employed to improve LLM reasoning (Xie et al., 2024a; Gao et al., 2024), and our implementation closely follows RAP (Hao et al., 2023), with key modifications. Specifically, we omit the use of a world model for predicting state changes and rely solely on the likelihood of each generated reasoning step as the intermediate reward, instead of LLM-based self-evaluation. Additionally, we introduce an oracle setting for MCTS, granting access to task rewards by comparing generated answers with ground truths during the search, providing an upper-bound performance benchmark.

MCTS inherently explores multiple reasoning chains by expanding nodes during the search process. We evaluate two result aggregation strategies: selecting the final answer from the chain with the highest trajectory reward and performing weighted voting over final answers based on their respective rewards. To ensure computational efficiency, we limit the tree's breadth to 4, consistent with our PGTS approach.

## E. PGTS Approach

The PGTS policy employs a Graph Neural Network (GNN) architecture to dynamically model the reasoning tree as the search progresses. Specifically, we utilize GPS layers (Rampášek et al., 2022), which integrate local message passing with global attention mechanisms. This design effectively captures both fine-grained structural relationships and broader contextual information within the reasoning process. The policy network consists of two GPS layers for feature aggregation, followed by a linear layer applied to the current node's features to predict both actions and values. The action prediction is modeled as a categorical distribution parameterized by the policy network, while the value prediction outputs a continuous scalar.

The input to the policy network is the reasoning tree constructed up to the current step. Each node in the tree represents a

partial reasoning step, with its features initialized using the embedding of the corresponding reasoning text. Edge features encode the immediate reward $R(s, a)$, which captures the contribution of each reasoning step to the overall trajectory.

Training the policy leverages Proximal Policy Optimization (PPO) (Schulman et al., 2017), a reinforcement learning algorithm designed to balance exploration and exploitation while maintaining stable updates. The training objective maximizes the expected cumulative reward across reasoning paths, with constraints to ensure updates remain within a stable region. Key PPO hyperparameters include:

- PPO Clip Range: 0.2

- Discount Factor ($\gamma$): 0.99

- GAE Lambda ($\lambda$): 0.95

During training, we use the current policy to simulate the reasoning trajectories and collect the partial reasoning trees. These trajectories are collected and used to optimize the policy based on PPO objectives, iteratively improving the efficiency and accuracy of reasoning over time.

## F. Alternative Policy Implementations

**SLM**   The Small Language Model (SLM) implementation substitutes the graph-based policy with a language-model-driven approach. We employ `distilbert-base-uncased`, a lightweight transformer model, to process the reasoning trajectory. The input to the model is the serialized reasoning chain corresponding to the current node. This serialized trajectory is passed through the model, which generates both the predicted next action and the value estimation using a linear output layer. During training, the backbone language model remains frozen, and only the output layer parameters are optimized. The optimization follows the same PPO objectives as used in PGTS.

**LLM Agent**   The LLM Agent represents an agentic approach where the same target LLM used for reasoning is also leveraged to predict the next exploration action. At each step, the reasoning trajectory is provided as a prompt to the LLM, along with instructions to suggest the next action. Please see Fig. 4 for the prompt template to select the exploration action. This approach aligns conceptually with a collaborative multi-agent framework where the LLM functions as both a generator to communicate with user and a planner to reason about the optimal action. However, the LLM Agent lacks explicit structural modeling of the reasoning process and relies heavily on prompt engineering to guide its predictions. While this approach demonstrates reasonable performance, it is limited by the inherent inefficiencies of treating the LLM as both the reasoning environment and the policy, leading to increased computational costs and suboptimal exploration in complex scenarios.

## G. Examples

Figure 5, 6, 7, and 8 illustrate examples of the PGTS reasoning process applied to problems from GSM8K, MATH, PrOntoQA, and StrategyQA, respectively. The left side displays the reasoning tree, while the right side outlines the detailed reasoning steps.

You are an AI assistant designed to navigate through a reasoning graph to solve problems. At each node in the graph, you can take one of these actions:

1. Expand: Dive deeper from current node to explore its subnodes
2. Branch: Explore alternative parallel paths by investigating sibling nodes
3. Backtrack: Return to a previous node to explore different possibilities
4. Terminate: End the reasoning process when you've reached a satisfactory answer

Important notes:
- Sibling nodes are generated dynamically when you choose to explore them
- Your goal is to find the most logical path to the correct answer
- Consider each step carefully before choosing an action
- Please choose only from the available actions list given below
- Choose 'Branch' when current node seem incorrect or inefficient towards a solution
- Choose 'Backtrack' to revise a previous node that may lead to incorrect or inefficient solution
- When 'Backtrack' to a previous node in the current path, specify its node id in the format 'Node Id: NODE_ID'
- Choose 'Terminate' only when you're confident in the complete solution

# QUESTION:
Here is the problem to be solved:
{question}

# CURRENT REASONING PATH:
Node IDs and reasoning steps:
{reasoning_path}

# AVAILABLE ACTIONS:
Due to graph constraints in depth and breadth, not all actions are available to choose at each step. The following actions are currently available:
{available_actions}

Instructions:
1. Analyze where you are in solving the problem
2. Review which actions are available to you
3. Decide your next action based on:
    - Is the current path promising?
    - Are there better approaches to try?
    - Have you reached a complete solution?
4. You can only choose one action from the available actions above
5. Format your final action choice as: <answer>YOUR_CHOSEN_ACTION</answer> and optionally provide your thinking process. 'YOUR_CHOSEN_ACTION' should be one of the available actions.

*Figure 4.* Prompt template to select the optimal exploration action for LLM Agent based policy.

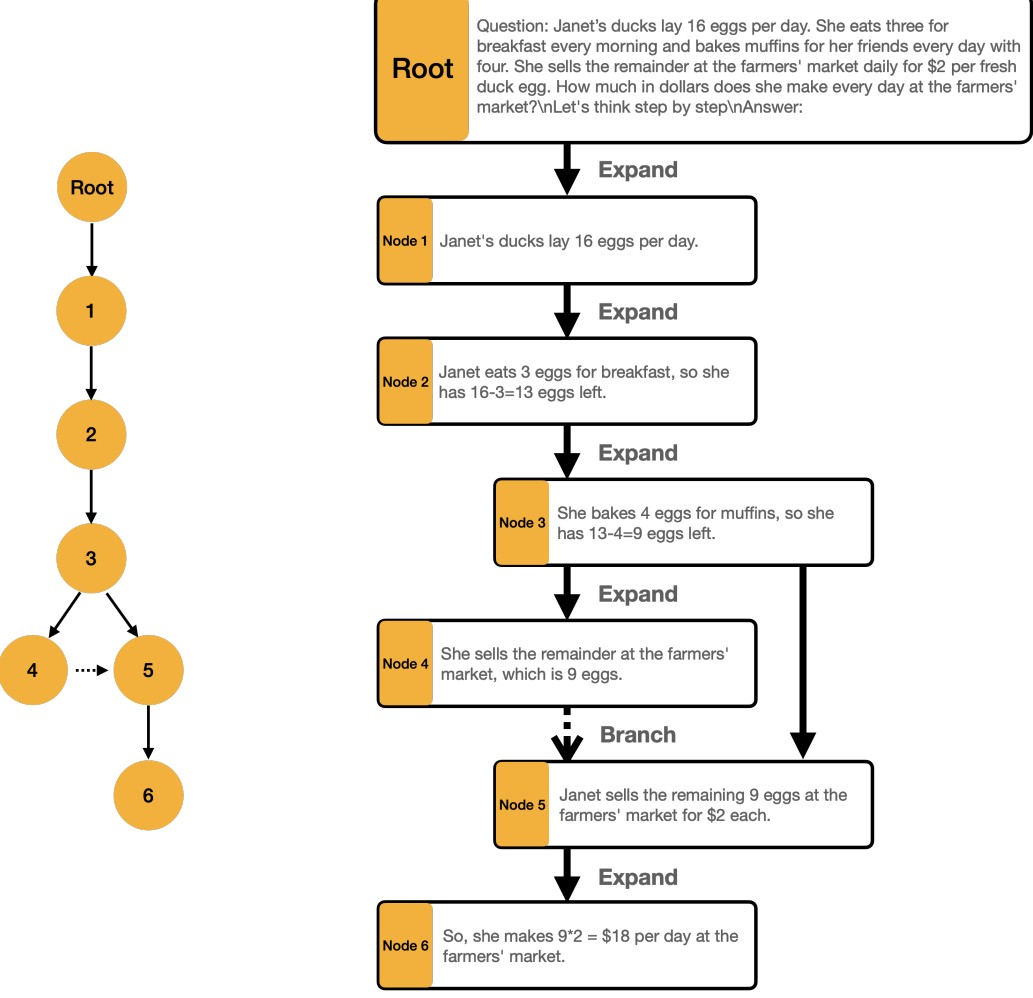

*Figure 5.* An example of the PGTS reasoning process applied to a problem from the GSM8K dataset. At node 4, the policy decide to branch to explore an alternative reasoning path.

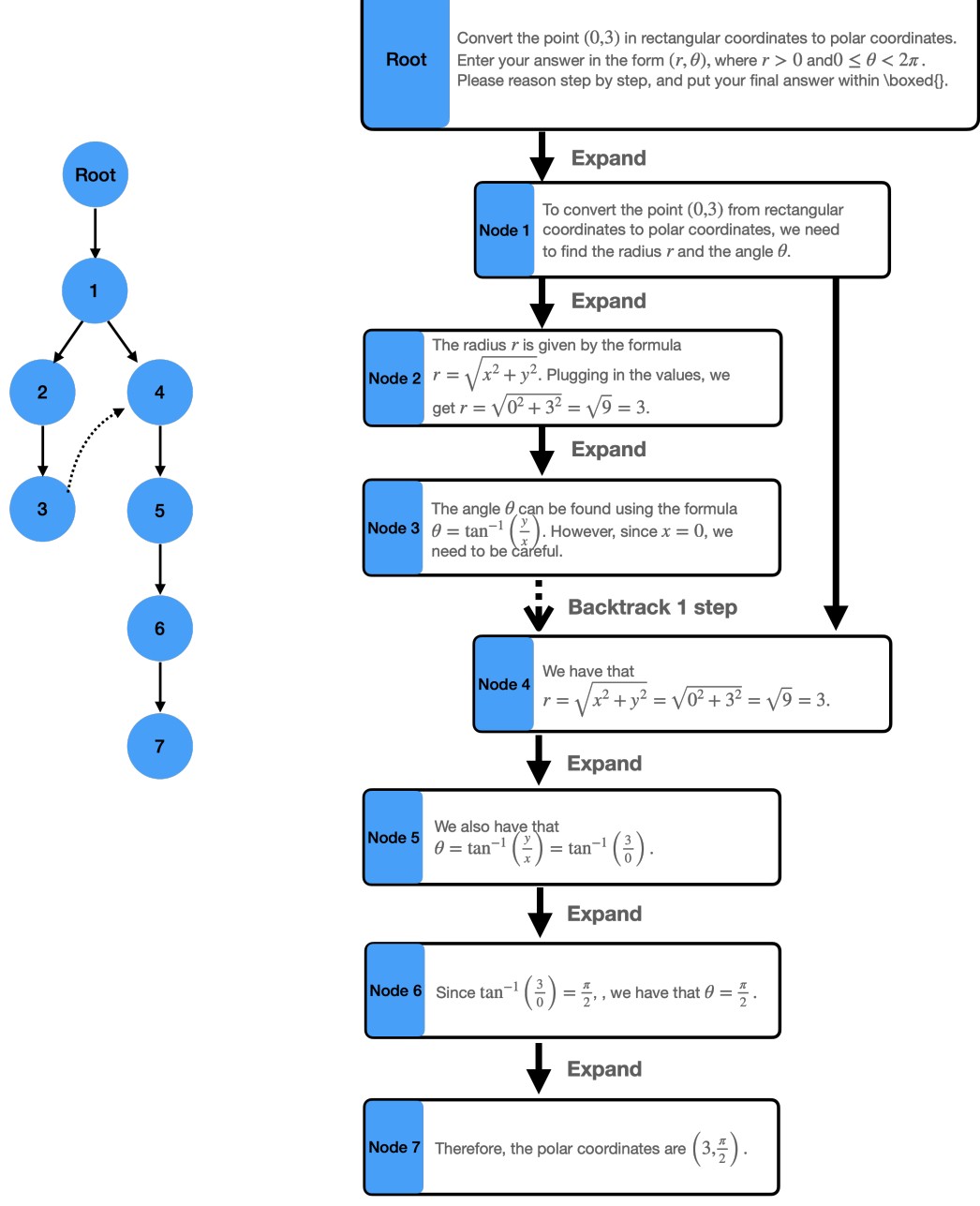

*Figure 6.* An example of the PGTS reasoning process applied to a problem from the MATH dataset. At node 3, the policy backtracks 1 step to explore an alternative reasoning path.

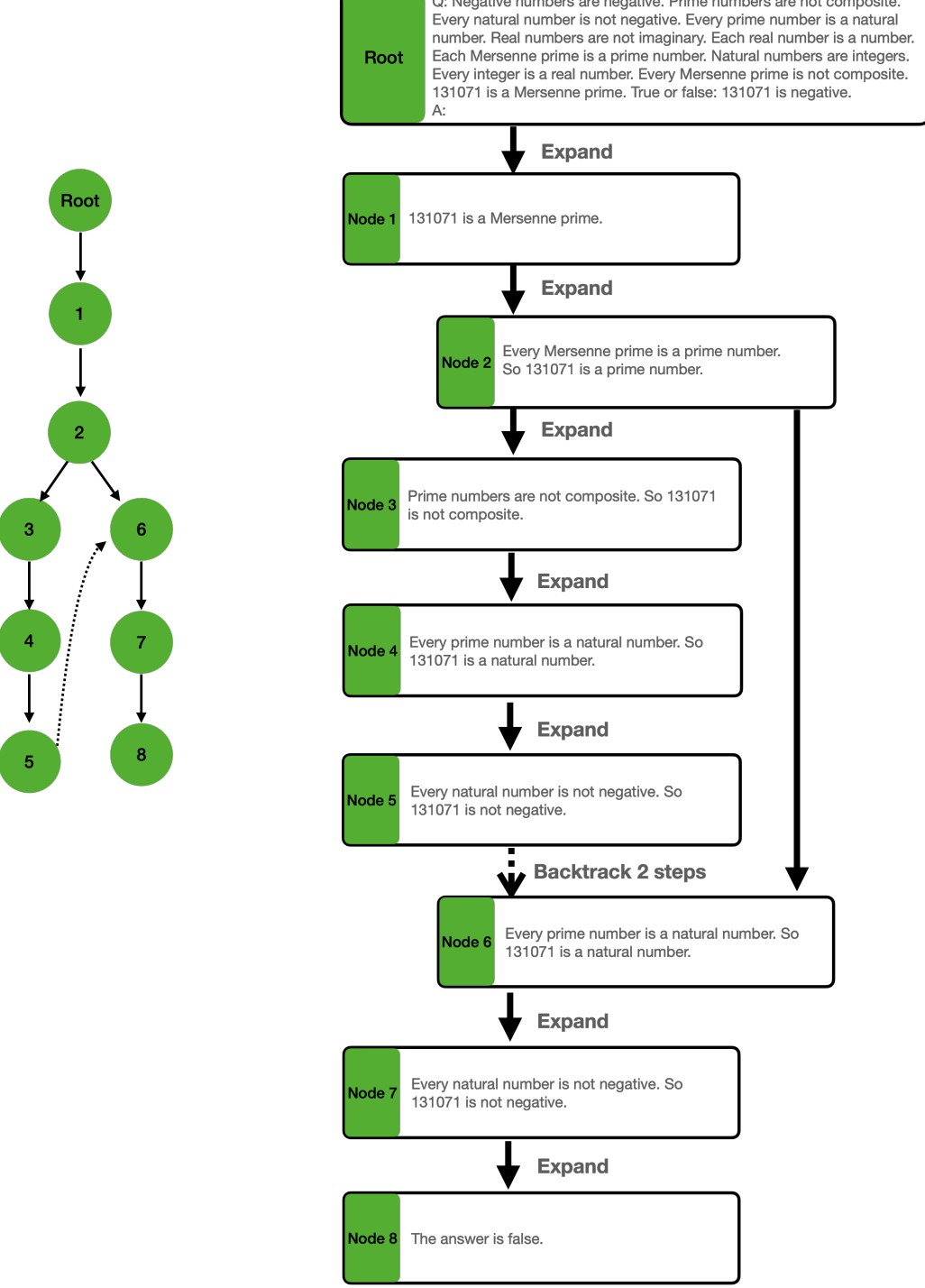

*Figure 7.* An example of the PGTS reasoning process applied to a problem from the PrOntoQA dataset. At node 5, the policy backtracks 2 steps to explore an alternative reasoning path.

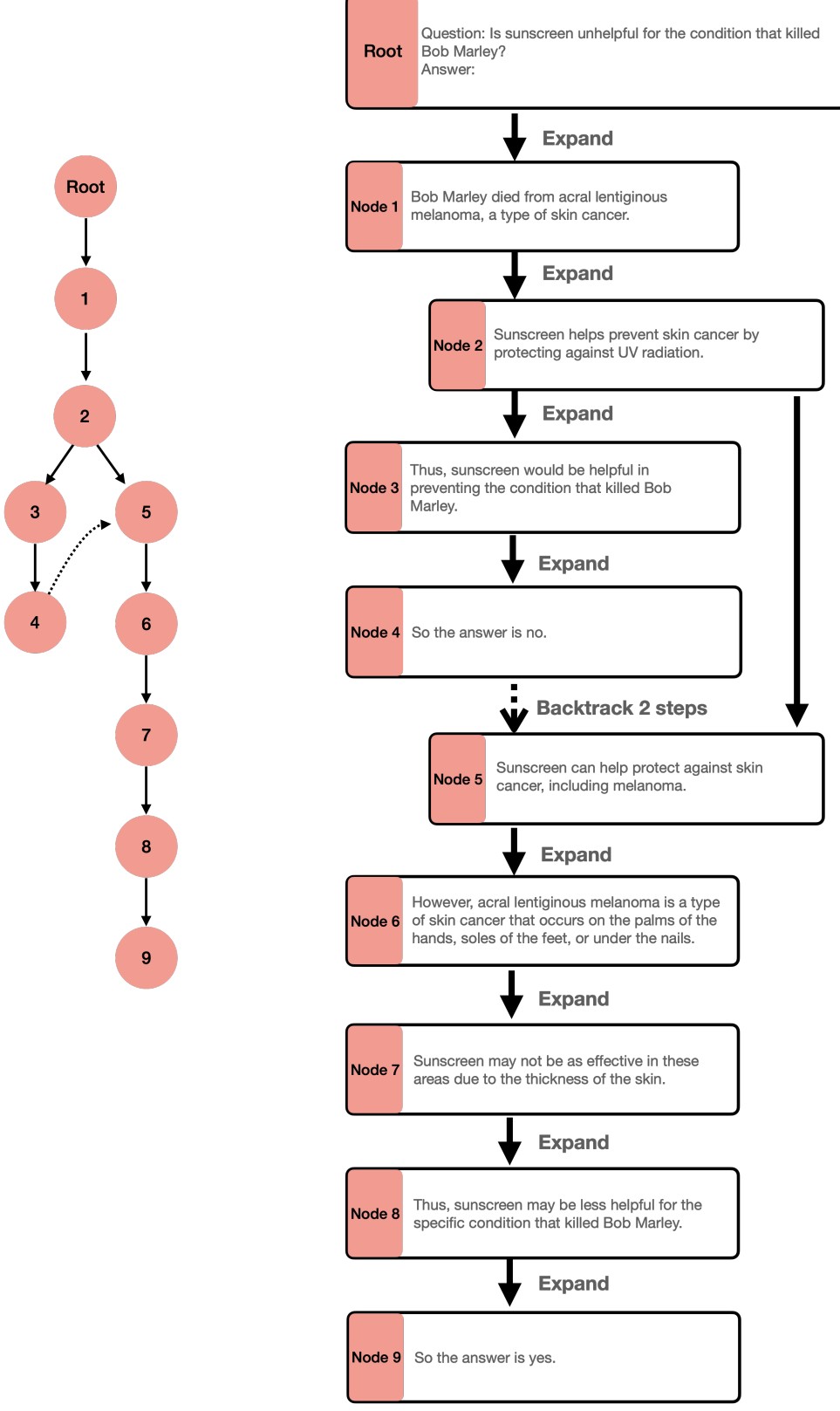

*Figure 8.* An example of the PGTS reasoning process applied to a problem from the StrategyQA dataset. Although node 4 has reached an answer, the policy chooses to explore alternative reasoning paths to refine and rectify the answer.

