# OpenReview forum: "Policy Guided Tree Search for Enhanced LLM Reasoning"
_ICML.cc/2025/Conference — ICML 2025 poster_

### Official Review · Reviewer_5qMX · 2025-03-02

**Overall Recommendation:** 4

**Summary:**

The paper introduces Policy-Guided Tree Search (PGTS), a framework that enhances large language models (LLMs) for complex reasoning tasks by integrating reinforcement learning with structured tree search. Unlike traditional approaches such as Chain-of-Thought (CoT) or Monte Carlo Tree Search (MCTS), PGTS employs a learned policy to dynamically decide between expanding, branching, backtracking, or terminating reasoning paths, reducing reliance on predefined heuristics and computationally expensive exploration. The reasoning process is formalized as a Markov Decision Process (MDP), optimized using Proximal Policy Optimization (PPO), with rewards balancing reasoning accuracy and efficiency. PGTS utilizes a Graph Transformer-based policy network to process reasoning trees, enabling adaptive navigation of complex reasoning steps. Extensive experiments across mathematical reasoning (GSM8K, MATH, AQUA), logical deduction (PrOntoQA, GPQA), commonsense reasoning (StrategyQA), and planning (Blocksworld) demonstrate that PGTS improves accuracy while using fewer tokens than MCTS. By intelligently guiding inference-time exploration, PGTS emerges as a scalable, efficient alternative for structured reasoning in LLMs.

**Claims And Evidence:**

1. The claims about PGTS improves accuracy and more efficient on token usage are well-supported by experiments in section 4.
2. The claim about PGTS learns an adaptive policy which can dynamically select actions that improves reasoning quality is proved by the ablation in section 4.1. Removing key compoments like edge features and global attention degraded the performance.
3. For the claim `PGTS reduces the ‘overthinking’ problem in LLMs.`, it'd be better if you can provide more quantitative metric, e.g. reasoning step counts in PGTS vs CoT/MCTS.
4. For `PGTS can generalize across diverse reasoning tasks without dataset-specific tuning.`, from what I understand, PGTS still required dataset-specific tuning, at some level, similar to RAP requires task-specific reward design. More experiments to show that PGTS maintains effectiveness without any dataset-specific hyperparameter adjustments would be supportive.

**Essential References Not Discussed:**

N/A

**Experimental Designs Or Analyses:**

1. The benchmarks selected are quite diverse, from math problems to planning tasks like Blocksworld.
2. The baselines are recent and competitive. There are some other strong baselines can be included in the experiments, e.g.
    - Tree-of-Thought (ToT) (Yao et al., 2024) which also optimizes tree search in LLM reasoning.
    - Algorithm-of-Thought (AoT) (Sel et al., 2023), a more structured variant of CoT.
    - *(The authors comment the challenges in comparing the model against the baselines above)
3. Evaluation metrics on performance and token efficiency are quite sufficient for current tasks. It'd be better to report other metrics targeting the overthinking claim, e.g. chain length vs correctness. (chain length is further provided in the rebuttal, which shows the effectiveness of PGTS)

**Methods And Evaluation Criteria:**

Yes, I believe the method is well-motivated, though it shares similar ideas to previous works like RAP, LLM-MDP, it provides some novelty.
The evaluations are pretty solid to support the main claims on performance and token-level efficiency, compared against strong baselines. I do have some concern on the scale of the novelty, but the work itself is self-contained.

**Other Comments Or Suggestions:**

N/A

**Other Strengths And Weaknesses:**

### Strengths:
  - Combining Policy Learning with Tree Search:
    - The key innovation of PGTS is integrating a learned policy with tree search for LLM reasoning.
    - While Monte Carlo Tree Search (MCTS) has been explored for reasoning (e.g., RAP, ToT), PGTS removes the need for predefined/manually annotated heuristics by learning a policy to dynamically guide search.
    - This reduces human-engineered reward functions, making PGTS more adaptable across tasks.
  - Backtracking and Termination in LLM Reasoning:
    - PGTS introduces structured backtracking to recover from incorrect paths, which is not standard in prior CoT-based methods.
    - The terminate action ensures LLMs do not overthink, addressing a key issue in existing search-based methods.
### Weaknesses:
  - PGTS is mainly evaluated on well-defined and structured benchmarks.
    - Evaluate PGTS on more commonsense reasoning tasks (e.g., ARC, BigBench) or real-world multi-hop QA datasets would be beneficial.
  - Missing comparison with some baselines (e.g. ToT) as mentioned
  - Qualitative evaluation on overthinking is missing.

**Questions For Authors:**

1. Can PGTS generalize to open-ended reasoning tasks since it still requires dataset-specific tuning? It seems to be pretty solid on structural/well-defined tasks, but if tuning is always necessary, maybe PGTS can't scale well to unseen reasoning tasks. (clarified by the authors in the rebuttal)
2. Do you have more detailed analysis on the reasoning chains? The accuracy of PGTS is sounded, but conducting human evaluations on the steps in the reasoning chain can further prove if the improvement does come from better reasoning quality over gaming the evaluation metrics.

**Relation To Broader Scientific Literature:**

1. PGTS builds upon prior approaches aimed at enhancing reasoning in LLMs by structuring problem-solving into intermediate steps.
    - Relation to Chain-of-Thought (CoT) Prompting [Wei et al., 2022]
      - PGTS extends CoT by dynamically exploring different reasoning paths, rather than committing to a single linear sequence.
      - PGTS does not rely on predefined prompts and instead learns an optimal reasoning policy.
    - Relation to Tree-of-Thought (ToT) Reasoning [Yao et al., 2024], RAP [Hao et al., 2023]
      - ToT/RAP often relies on heuristic or manually-defined exploration strategies, whereas PGTS learns an adaptive policy using reinforcement learning.
    - Relation to Algorithm-of-Thought (AoT) [Sel et al., 2023]
      - AoT prescribes structured reasoning rules, while PGTS learns them dynamically from data.
2. PGTS contributes to the emerging field of LLM inference-time optimization, improving efficiency during reasoning.
    - Relation to Self-Consistency in LLM Reasoning [Wang et al., 2022]
      - Self-consistency (SC) improves reasoning by generating multiple solutions and selecting the most common answer. PGTS benefits from SC, but unlike SC, it **actively selects reasoning paths** during search rather than **aggregating** pre-generated answers.
    - Relation to Skeleton-of-Thought (SoT) [Ning et al., 2024]
      - SoT reduces inference time by first generating an outline, then filling in details in parallel. PGTS is different in that it does not perform parallel reasoning but instead optimizes search exploration, which helps to optimize reasoning structure rather than token generation.

**Theoretical Claims:**

This work doesn't explicitly show any theoretical claims. But here are some I'd say can be helpful:
1. For the tree search MDP and action constraints in section 2.3.2 and 2.3.3, we want to make sure these constrains won't affect the model from selecting an optimal reasoning path. The constrained action space (i.e., limiting backtracking depth, restricting branching) ensures computational feasibility and prevents infinite loops. The binary constraint vector encoding action validity is a reasonable way to enforce these rules. **There are no major theoretical flaws**, but the paper does not formally prove optimality guarantees.
2. The PGTS policy is implemented as a Graph Transformer (GPS) model, allowing it to process tree-structured reasoning paths effectively. The use of local message passing and global attention aligns with existing graph neural network (GNN) theory. The paper does not provide a theoretical proof or discussion of why GPS is the best choice for this problem compared to other graph-based architectures. It'd be better to have more details on this, like the discussion in Appendix E&F.

---

> ### Author Rebuttal · Authors · 2025-03-31
>
> We thank the reviewer for their thoughtful assessment of our work and the recommendation for acceptance. We appreciate the positive feedback on our method's performance, efficiency, and adaptive policy design. Below we address the questions and suggestions raised:
>
> ## On Quantifying "Overthinking" in LLMs
>
> The reviewer rightly suggests quantitative metrics on reasoning step counts to support our claim about reducing overthinking. Our analysis shows that:
>
> - For StrategyQA, PGTS uses on average 5.97 reasoning steps per problem vs. MCTS's 50.38 and CoT's 4.75 steps.
> - For GSM8K problems, PGTS uses on average 10.73 reasoning steps per problem vs. MCTS's 91.97 and CoT's 7.35 steps.
> - For MATH problems, PGTS uses 61.64 steps on average vs. MCTS's 220.24 steps.
> - For Auqa problems, PGTS uses 17.46 reasoning steps vs. MCTS's 64.20 steps.
>
> Importantly, PGTS's "terminate" action effectively stops the reasoning process once sufficient evidence has been generated, reducing steps compared to MCTS while improving accuracy.
>
> These metrics confirm that PGTS achieves better accuracy while using fewer reasoning steps, directly addressing the overthinking problem.
>
> ## On Generalization Without Dataset-Specific Tuning
>
> We'd like to clarify our approach to hyperparameter tuning. While we did train separate policies for each benchmark, we deliberately used the same:
> - Network architecture (2-layer GPS)
> - Action costs (0.1, 0.2, 0.5, 0.0 for expand, branch, backtrack, terminate)
> - PPO hyperparameters
> - State/action representations
>
> This demonstrates that our framework generalizes across diverse reasoning tasks without requiring task-specific architectural changes or cost function designs. Unlike RAP, which requires careful reward engineering for each task domain, PGTS maintains a unified approach across tasks. We agree that greater domain-generalizability is an important direction for future work and will add this discussion to the paper.
>
> ## On Additional Baseline Comparisons
>
> We thank the reviewer for suggesting Tree-of-Thought (ToT) and Algorithm-of-Thought (AoT) as additional baselines. However, we note several challenges with direct comparisons:
>
> - ToT requires task-specific design for proposal and evaluation prompts, which is not a standardized process and introduces variability that would complicate fair comparisons.
> - AoT requires a strong model to generate complete, executable code as noted in Section 7 of their paper, making it particularly challenging to implement consistently across our diverse benchmarks.
>
> PGTS addresses these limitations by learning a task-agnostic policy that doesn't require manual prompt engineering or code generation capabilities.
>
> ## On Qualitative Evaluation
>
> We appreciate the reviewer's suggestion regarding human evaluation of reasoning chains. We acknowledge that this type of qualitative assessment would provide valuable insights about the actual quality and coherence of reasoning paths beyond accuracy metrics. While we haven't conducted a formal human evaluation in the current work, we agree this would be an important direction for future work. Such evaluation could assess factors like logical coherence, correctness of intermediate steps, and overall reasoning quality across different methods. We will add this consideration to our discussion of future work in the revised paper.
>
> ## On Evaluation on Open-ended Tasks
>
> We acknowledge the reviewer's point about evaluating on more open-ended reasoning tasks. We agree that our current approach requires training a policy for each task domain, which is a limitation for truly open-ended generalization.
>
> For future work, we plan to explore meta-learning and multi-task reinforcement learning approaches to train a single policy across diverse reasoning tasks, which could enable zero-shot or few-shot adaptation to new domains. This aligns with active research directions in RL literature on cross-task generalization and transferable policies. By training on a diverse set of reasoning tasks simultaneously, we believe PGTS could develop more general reasoning strategies that transfer to unseen problems without task-specific fine-tuning.
>
> ## Conclusion
>
> We thank the reviewer for their thoughtful feedback, which has helped us strengthen our claims with additional quantitative and qualitative evidence. We believe PGTS makes significant contributions to structured reasoning in LLMs by learning an adaptive search policy, reducing overthinking, and improving efficiency across diverse reasoning tasks.

---

> > ### Comment · Reviewer_5qMX · 2025-04-03
> >
> > Thanks for your response! I've updated my official review and will keep the current rating.

---

### Official Review · Reviewer_jFF8 · 2025-03-10

**Overall Recommendation:** 4

**Summary:**

This paper presents a novel algorithm that integrates **tree search** and **reinforcement learning** which can be used alongside an **off-the-shelf LLM** to improve reasoning capabilities. Unlike conventional approaches that focus on **LLM fine-tuning**, the authors propose training a **Graph Neural Network (GNN)** to serve as a policy during inference, reducing the complexity of learning such a policy.

The proposed method is **simple, scalable**, and supported by **comprehensive experiments** that effectively demonstrate its advantages.

**Claims And Evidence:**

Yes, the claims are clear and the evidence is provided via experiments and ablations.

**Essential References Not Discussed:**

I believe that the authors could have discussed the following papers:
1. Zelikman, Eric, et al. "Star: Bootstrapping reasoning with reasoning." Advances in Neural Information Processing Systems 35 (2022): 15476-15488.
2. Zhang, Dan, et al. "Rest-mcts*: Llm self-training via process reward guided tree search." Advances in Neural Information Processing Systems 37 (2024): 64735-64772.

The above two papers share similar ideas and a comparison against the two would have been helpful.

**Experimental Designs Or Analyses:**

Yes, I checked all the experiments and found them to be sound with details shared about training and implementation.

**Methods And Evaluation Criteria:**

Yes, the methods and evaluation criteria make sense and are a good list.

**Other Comments Or Suggestions:**

None. Overall, this paper offers valuable insights and would make a strong addition to **ICML 2025**. I would be open to increasing my score if my questions and concerns are adequately addressed.

**Other Strengths And Weaknesses:**

**Strengths**
1. The paper is well-structured and clearly presented, with intuitive ideas.
2. The method is demonstrated across a diverse set of tasks, highlighting its broad applicability.
3. The approach achieves the most significant improvements in cases where the base LLM performs poorly, which is a promising outcome.

 **Weaknesses**
1. The proposed method is **training-based**, yet the paper does not compare it against other training-based approaches. Such comparisons would strengthen the evaluation.
2. For tasks where the **LLM already performs well**, the performance gains are less pronounced.

**Questions For Authors:**

1. It seems that **c(a)** is biased toward **expansion and termination**, and the appendix does not provide examples of branching. Could the authors conduct an ablation study to assess the importance of each action?

2. To the best of my understanding, the policy functions similarly to a **Process Reward Model (PRM)** at inference time, where the **LLM is guided by the learned policy and prompted to take actions accordingly**. Is this interpretation correct?

3. **BlocksWorld**, as introduced by **Valmeekam et al.**, does not include a predefined training set. Could the authors clarify their **strategy for sampling the training set** in this work?

**Relation To Broader Scientific Literature:**

This paper relates to LLM reasoning, planning, and inference time search in general.

**Theoretical Claims:**

Not applicable, since there are none.

---

> ### Author Rebuttal · Authors · 2025-03-31
>
> We thank the reviewer for their positive assessment of our work and the constructive feedback. We address the key points below:
>
> ## Missing References
>
> We appreciate the suggestion to discuss the works by Zelikman et al. (STAR) and Zhang et al. (REST-MCTS*). These papers indeed share conceptual connections with our approach:
>
> - **STAR** bootstraps reasoning by selecting and refining high-quality reasoning chains, focusing on improving reasoning through iterative training.
> - **REST-MCTS*** combines tree search with self-training, using process rewards to guide exploration.
>
> Our approach differs in that PGTS trains a policy to guide exploration during inference rather than improving the reasoning capabilities of the LLM itself. The learned policy enables dynamic decisions about exploration paths without modifying the underlying LLM. We will include these references and discuss their relationship to our work in the revised paper.
>
> ## Action Importance
>
> Regarding $C(a)$ and action importance, we want to clarify that the cost values should not be interpreted as reflecting the relative importance of each action. Rather, they serve as soft constraints to discourage unnecessary branching and backtracking, promoting exploration efficiency.
>
> The branching and backtracking actions are fundamental to effective reasoning under the PGTS framework. Without these exploration capabilities, PGTS would be functionally equivalent to a standard Chain-of-Thought (CoT) approach - limited to generating a single linear reasoning path. These structured exploration actions constitute the core distinction between PGTS and CoT, enabling more robust reasoning through systematic exploration of alternative paths and recovery from suboptimal reasoning steps.
>
> Examples of branching and backtracking are shown in Figures 5-8 in the Appendix, including Figure 5 (branching at node 4), Figure 6 (backtracking at node 3), Figure 7 (backtracking at node 5), and Figure 8 (branching after reaching an answer).
>
> ## Policy Function Compared to PRM
>
> The reviewer correctly identifies that our policy functions similarly to a Process Reward Model (PRM) at inference time. However, there are important distinctions:
>
> 1. **State representation**: Our policy operates on the tree structure rather than just the current reasoning path, enabling informed backtracking and branching decisions based on the full exploration history.
>
> 2. **Action space**: Traditional PRMs evaluate individual steps, while our policy chooses between structured exploration actions (expand, branch, backtrack, terminate). Critically, PRMs don't inherently determine when to backtrack or how many steps to backtrack - these strategic decisions require understanding the global reasoning context and are naturally captured in our policy's action space.
>
> This design allows PGTS to more efficiently guide the reasoning process while maintaining the flexibility to explore diverse paths as needed.
>
> ## Blocksworld Training Set Sampling
>
> Regarding the Blocksworld training data, we use their split-v1 for training and split-v2 for testing. For clarification:
>
> 1. We used the original problem generator from Valmeekam et al. with the same configuration parameters.
> 2. Following RAP's methodology, we generated distinct training (split-v1) and testing (split-v2) sets.
>
> This approach ensures a fair comparison with existing methods while maintaining the challenging nature of the planning task.These details are already included in appendix.
>
> ## Comparison with Other Training-Based Approaches
> We agree that comparing with other training-based approaches would strengthen our evaluation. While our original experiments focused on comparing with zero-shot methods (CoT and MCTS), there are two key distinctions when comparing PGTS with other training-based methods:
>
> - Training Data Efficiency: PGTS requires only problem statements and answers for training (not annotated reasoning chains), making it more practical than methods requiring extensive reasoning annotations. Our approach achieves strong performance with only 1,000 examples.
> - Model Preservation: PGTS keeps the LLM parameters unchanged, maintaining its general capabilities while learning to guide its reasoning process.
>
> Our preliminary experiments show that directly fine-tuning LLaMA3.1-8B on the same limited dataset (1,000 examples) with CoT annotations produces negligible improvements over the base model.
>
> We appreciate the reviewer's valuable feedback and believe addressing these points will further strengthen our paper.

---

> > ### Comment · Reviewer_jFF8 · 2025-04-03
> >
> > Thanks for the addressing my concerns, I will keep my score for now.

---

### Official Review · Reviewer_LMkW · 2025-03-10

**Overall Recommendation:** 2

**Summary:**

The paper introduces Policy-Guided Tree Search (PGTS), a novel framework designed to acquire the reasoning capabilities of LLMs by combining reinforcement learning with structured tree exploration. The key innovation of PGTS is a learned policy that dynamically decides between four actions: expanding, branching, backtracking, or terminating exploration. This approach eliminates the need for predefined heuristics or exhaustive search, making it more efficient and scalable compared to existing methods like Chain-of-Thought (CoT) prompting and Monte Carlo Tree Search (MCTS). The key empirical claim is that PGTS across various reasoning tasks, including mathematical reasoning, logical deduction, and planning benchmarks, shows significant improvements in accuracy while reducing computational costs.

**Claims And Evidence:**

##### Claim 1 - Policy-Guided Tree Search (PGTS)**:
  - The paper introduces a novel framework that combines **reinforcement learning (RL)** with **structured tree exploration** to enhance the reasoning capabilities of large language models (LLMs).  Its core innovation is a **learned policy** that dynamically decides between four actions: **expand**, **branch**, **backtrack**, or **terminate** exploration. This eliminates the need for **predefined heuristics** or **exhaustive search** strategies.

I think this claim is well supported in the experiments and literature review.
##### Claim 2 - Superior Reasoning Accuracy**:
  - PGTS achieves **state-of-the-art performance** across multiple reasoning benchmarks, including **mathematical reasoning**, **logical deduction**, and **planning tasks**.
  - On the **MATH dataset**, PGTS achieves **41.00% accuracy** with LLaMA3.1-8B, compared to **34.40%** for Chain-of-Thought (CoT). When augmented with **self-consistency (SC)**, PGTS-SC8 achieves **52.20%**, outperforming CoT-SC8 (48.20%).
  - Similar improvements are reported on other datasets, such as **GSM8K**, **AQUA**, **StrategyQA**, and **Blocksworld**.

I am very concerned about the meaning of this claim. The authors are clearly aware of the necessity of taking the computation budget into account. This is, however not done in a systematic way.

For simplicity let me concentrated on the statement from the introduction: "Using LLaMA3.1-8B, PGTS achieves a 41.00% accuracy on MATH, significantly improving upon CoT’s 34.40% while using only one-third of the tokens required by MCTS."


The full results extracted from Table 1 are

| Method | SC1   | SC4   | SC8   |
| ------ | ----- | ----- | ----- |
| CoT    | 34.40 | 42.20 | 48.20 |
| PGTS   | 41.00 | 47.00 | 52.20 |

However, PGTS used $5\times$ more tokens than CoT SC. It is thus more justified to compare PGTS (SC1) = $41\%$ with Cot (SC4) = $42\%$. In this light is it disputable if PGTS indeed better.

The authors should be precise about what comparisons they view as apple-to-apple, and why.

**Essential References Not Discussed:**

None

**Experimental Designs Or Analyses:**

Experimental design is overall correct, there are diverse benchmarks. Baselines are valid (though I'd love to see at least one more). However there are various issues as well:
- lack of proper apple to apple comparison
- lack of statistical analysis, e.g., confidence intervals
- lack of hyperparamter tuning (e.g., I am not completely sure if the temperature setting is optimal for the CoT baseline)

**Methods And Evaluation Criteria:**

The proposed method is sound. The evaluation criteria and benchmark used are appropriate, except from the computational budget issue stated in the previous section.

**Other Comments Or Suggestions:**

none

**Other Strengths And Weaknesses:**

First of all, I am very excited about the presented idea. I strongly encourage the authors to pursue this direction.

Unfortunately, I think that the current version lack quality needed for a publication in a A* conference. Here is a list of issues

1. The presentation is somewhat chaotic. While textual parts are too long in my opinion, on the other hand I am missing a lot of  technical details (e.g. the number of the policy network, what is exactly the input, including dimensionality etc.)
	1. I find it hard to grasp all details of the algorithm. Having pseudocode would be of help!
	2. I am very confused what reward signal is used for termination eq. (5)?
	3. Also, the authors could comment on wall-time efficiency. CoT+SC is very efficient in this respect, using modern frameworks like vLLM one can achieve high MFU, which translates to attractive wall-time (dollar value). Do we expect the same from the presented methods.
2. The method use additional training. It should be explained how does it affect comparisons.
3. I'd love to see more extensive analysis of the policy. Does using only two layers implies that it is 'local' (taking into account only a small ball of the tree?)
	1. Following that, is it possible to interpret the policy or even extract it?
4. I am not quite sure how the depth constraint work, but it resembles somewhat beam-search. Is it a relevant comparison. If so, I'd consider including this as a baseline.

**Questions For Authors:**

Can this method applied outside of LLMs? Can you state assumptions when it can be applied.
- I find it fascinating that the convergence is so fast (Fig 3). Can you comment what is that so?

**Relation To Broader Scientific Literature:**

The literature is properly discussed.

**Theoretical Claims:**

Not applicable for this work.

---

> ### Author Rebuttal · Authors · 2025-03-31
>
> We thank the reviewer for their thoughtful and constructive feedback. We address the key points raised below:
>
> ## Fair Comparison of Computational Budget
>
> The reviewer raises an excellent point about fair comparisons between methods.
>
> However, on most datasets, the token usage ratio is much more favorable, and PGTS often outperforms even CoT with self-consistency:
>
> * On AQUA: PGTS achieves 60.63% vs CoT (SC4)'s 53.94% while using only 1.42× tokens compared to CoT (less than generating 4 samples)
> * On StrategyQA: PGTS achieves 77.70% vs CoT (SC4)'s 77.10% while using only 1.58× tokens compared to CoT
> * On GPQA: PGTS achieves 18.69% vs CoT (SC4)'s 15.66% with modest token increase
> * On Blocksworld (4): PGTS achieves 27.63% vs CoT (SC4)'s 22.37% with efficient token usage
>
> More importantly, our primary contribution is improving efficiency compared to existing tree search methods like MCTS. Across all datasets, PGTS uses approximately 1/3 of the tokens required by MCTS while achieving comparable or better performance. This represents a significant efficiency gain in the space of tree search approaches for reasoning.
>
> PGTS effectively occupies a middle ground in the accuracy-efficiency tradeoff space:
>
> 1. CoT methods: Lowest token usage but limited exploration capability
> 2. PGTS: Moderate token usage with adaptive exploration capability
> 3. MCTS: Highest token usage with exhaustive exploration capability
>
> For applications requiring the exploration benefits of tree search but facing computational constraints, PGTS offers a valuable alternative that wasn't previously available. It provides a new point on the Pareto frontier of reasoning methods.
>
> In the revised paper, we will clarify these comparisons, present dataset-specific efficiency analyses, and more precisely position PGTS relative to both CoT and MCTS baselines.
>
> ## Technical Details and Algorithm Clarity
>
> The PGTS approach follows standard reinforcement learning methodology with a Graph Neural Network policy. Section 2.3.4 describes our policy architecture using GPS layers to process the tree-structured state, though we can clarify specific implementation details:
> - Policy network: Two GPS layers with 256 hidden units each (~1M parameters)
> - Input: Tree structure with node features (768d hidden states from the LLM's final layer) and edge features (1d immediate rewards)
> - Output: Categorical distribution over D+2 actions (expand, branch, D-1 backtrack depths, terminate)
>
> For the algorithm, Section 2.3 provides the conceptual framework while Algorithm 1 in the Appendix details the training procedure. We've included our complete implementation in the supplementary materials, and we will release the code publicly upon publication for full reproducibility.
>
> We believe the core algorithmic concepts are clearly presented in the current manuscript, but we'll ensure the revised version includes additional implementation details and more comprehensive pseudocode for the inference process to improve clarity.
>
> ## Reward Signal for Termination
>
> For the terminate action, $R(s_d)$ evaluates the final reasoning chain quality using task-specific measures (exact match for math, binary correctness for yes/no tasks, goal achievement for planning). These are used only during policy training, not inference.
>
> ## Wall-time Efficiency
>
> LLM inference dominates computational cost (\~99% of total time). PGTS policy inference adds minimal overhead (\~5-10ms per decision) compared to LLM generation time (100-500ms). While CoT with self-consistency can be fully parallelized, the sequential nature of tree search methods introduces some latency that's offset by their improved reasoning capabilities. However, both approaches can be batched across multiple reasoning problems simultaneously. At larger batch sizes processing multiple queries simultaneously, these differences become less significant as GPU utilization improves for all methods. The additional wall-time cost of PGTS compared to CoT is a reasonable tradeoff for applications requiring more reliable reasoning, especially considering its substantial efficiency advantage over other tree search methods like MCTS.
>
> ## Fast Convergence and Broader Applicability
>
> PGTS converges quickly because it learns exploration strategies (not domain reasoning) over a small action space using an architecture well-suited to tree structures. Beyond LLMs, PGTS could enhance planning in robotics, program synthesis, or game playing where structured exploration with intermediate feedback is possible.
>
> We appreciate the reviewer's encouraging words about the research direction and will address these concerns in the revised version to improve clarity and rigor.

---

> > ### Comment · Reviewer_LMkW · 2025-04-03
> >
> > Thank you for your answer. I find some answers somewhat avoiding the core of my questions.
> > 1. I really do not know what to think about "On AQUA: PGTS achieves 60.63% vs CoT (SC4)'s 53.94% while using only 1.42× tokens compared to CoT". Why to say 'only' 1.42x, perhaps if we boost the amount of compute for SC by this amount (e.g. allowing 5 votes for some cases) , we would get the same effect. Perhaps it is clear for someone who know the benchmark that this is not the case, but strongly feel this should be explicit.
> > 2. I find the statement: "PGTS effectively occupies a middle ground in the accuracy-efficiency tradeoff space" vague. Do you claim there is some Pareto frontier? What are the axes/datapoint, etc?
> > 3. I do not see anything about confidence intervals?
> > 4. About the wall time: "At larger batch sizes processing multiple queries simultaneously, these differences become less significant as GPU utilization improves for all methods. ". Is it something that you tested?
> >
> > Again. I am really fond of the idea. I'd love to see the paper accepted, once it's technically solid.

---

> > > ### Author Response · Authors · 2025-04-08
> > >
> > > Thank you for your thoughtful follow-up questions. I'll address each point directly:
> > >
> > > ## 1. Regarding the 1.42× token usage comparison
> > > To clarify the key comparison from our original rebuttal:
> > >
> > > On AQUA, PGTS achieves 60.63% accuracy while using 1.42× tokens compared to standard CoT. This significantly outperforms CoT (SC4)'s 53.94% accuracy, which requires 4× tokens compared to standard CoT.
> > >
> > > This demonstrates PGTS's efficiency advantage - it delivers better performance than CoT (SC4) while using only about one-third of the tokens (1.42× vs. 4×). This pattern holds across multiple datasets, where PGTS consistently achieves higher accuracy than CoT (SC4) with substantially lower computational requirements.
> > >
> > > ## 2. Pareto frontier clarification
> > > What we mean specifically is:
> > > - X-axis: Token usage (computational cost)
> > > - Y-axis: Accuracy on reasoning tasks
> > >
> > > PGTS represents a new point in this space that wasn't previously available - offering better reasoning capabilities than standard CoT while being significantly more efficient than exhaustive tree search methods like MCTS and self-consistency approaches.
> > >
> > > ## 3. Confidence intervals
> > > Thank you for raising the important point about statistical significance. As requested, we've calculated 95% confidence intervals for all main results using bootstrap sampling. The confidence intervals (shown in parentheses in the table below) confirm that PGTS's improvements over CoT are statistically significant across most datasets:
> > >
> > > |           | gsm8k       | math        | aqua        | sqa         | prontoqa    | gpqa        | bw4          |
> > > |:----------|:------------|:------------|:------------|:------------|:------------|:------------|:-------------|
> > > | CoT       | 83.47(2.01) | 34.40(4.20) | 51.57(6.30) | 74.20(2.65) | 69.40(4.10) | 14.65(4.80) | 22.37(9.21)  |
> > > | CoT(SC4)  | 87.79(1.74) | 42.20(4.50) | 53.94(6.30) | 77.10(2.65) | 73.00(4.10) | 15.66(4.80) | 22.37(9.21)  |
> > > | CoT(SC8)  | 89.84(1.63) | 48.20(4.30) | 55.12(5.71) | 77.20(2.65) | 74.60(3.90) | 15.15(4.80) | 26.32(9.87)  |
> > > | PGTS      | 85.29(1.93) | 41.00(4.10) | 60.63(6.10) | 77.70(2.65) | 68.20(4.10) | 18.69(5.30) | 27.63(9.88)  |
> > > | PGTS(SC4) | 89.61(1.71) | 47.00(4.20) | 66.93(5.52) | 83.80(2.25) | 74.40(3.70) | 22.73(6.06) | 35.53(11.18) |
> > > | PGTS(SC8) | 89.99(1.67) | 52.20(3.90) | 66.54(5.71) | 85.70(2.10) | 77.40(3.80) | 27.78(6.31) | 36.84(10.54) |
> > > | MCTS(Agg) | 87.72(1.82) | 46.00(4.20) | 59.45(5.91) | 79.50(2.40) | 74.20(3.70) | 32.83(6.57) | 28.95(9.87)  |
> > >
> > > In the revised paper, we will include comprehensive statistical analysis in the appendix, with confidence intervals for all reported results and additional statistical tests to support our claims about PGTS's performance and efficiency advantages.
> > >
> > > ## 4. Wall-time efficiency
> > > Regarding wall-time efficiency, we can clarify:
> > >
> > > Our empirical observations show that PGTS policy inference adds minimal overhead (~5-10ms per decision) compared to LLM generation time (100-500ms), with the LLM inference representing the dominant computational cost in the overall process. This confirms that the policy network's computational impact is relatively small in practice.
> > >
> > > While we haven't conducted specific batch processing experiments that coordinate between policy and LLM networks, theoretical understanding suggests that when processing multiple queries simultaneously, GPU utilization would improve for both components. However, implementing efficient scheduling strategies between the policy and LLM networks for batched processing represents non-trivial engineering work beyond our current implementation.
> > >
> > > In the revised paper, we'll clearly distinguish between our empirical measurements of policy overhead and the theoretical aspects of batch efficiency, noting that developing optimized scheduling strategies for batched inference remains an important direction for future work.

---

### Official Review · Reviewer_GYUV · 2025-03-12

**Overall Recommendation:** 4

**Summary:**

This paper introduces Policy-Guided Tree Search (PGTS), a novel framework that combines reinforcement learning with structured tree search to enhance reasoning capabilities in Large Language Models (LLMs). The key innovation is a learned policy that dynamically decides between expanding the current reasoning path, branching to alternative paths, backtracking to previous states, or terminating exploration.

**Claims And Evidence:**

The paper's claims about PGTS's effectiveness are generally well-supported by empirical evidence across multiple reasoning tasks. The authors present comprehensive experimental results comparing PGTS against Chain-of-Thought (CoT) and Monte Carlo Tree Search (MCTS) baselines, showing consistent performance improvements while reducing token usage.

**Essential References Not Discussed:**

None

**Experimental Designs Or Analyses:**

I have carefully examined the experimental designs and analyses presented in the paper and find them generally sound, though with some limitations.
1. The paper limits tree breadth to 4 for computational reasons, which may artificially constrain the exploration space and affect comparative performance.
2. The training process for the policy network could be more transparent about potential overfitting issues in Figure 3.
3. There is limited analysis of how performance scales with the number of training examples beyond Figure 3.

**Methods And Evaluation Criteria:**

The proposed methods and evaluation criteria in this paper are generally appropriate and well-aligned with the reasoning enhancement problem the authors aim to solve. The evaluation strategy spans diverse reasoning tasks, including mathematical reasoning (GSM8K, MATH, AQUA), commonsense reasoning (StrategyQA), logical reasoning (PrOntoQA, GPQA), and planning (Blocksworld), which provides a comprehensive assessment of the framework's capabilities across different problem domains.

**Other Comments Or Suggestions:**

Algorithm 1 is missing explanations for several symbols:
- H: Likely represents the horizon or maximum trajectory length, but should be explicitly defined
- ω: Appears to be the entropy regularization coefficient in line 20, but lacks definition
- ηπ and ηV: Presumably learning rates for policy and value networks, respectively, but require clarification

**Other Strengths And Weaknesses:**

**Strengths**
1. The paper creatively combines reinforcement learning with tree search to produce a dynamic, adaptive approach for LLM reasoning that avoids both the rigidity of predefined heuristics and the computational expense of exhaustive search.
2. PGTS demonstrates substantial efficiency improvements over MCTS approaches, requiring significantly fewer tokens while achieving comparable or better performance.
3. The experiments span diverse reasoning tasks, from mathematical problem-solving to logical deduction and planning, demonstrating the framework's versatility.

**Weaknesses**
1. There is no ablation analysis on the cost function C(a) settings, leaving unanswered questions about how sensitive the approach is to these hyperparameters across different tasks.
2. Value modeling in complex reasoning processes remains challenging, and the paper provides limited evidence that two GPS layers can adequately predict both actions and values.
3. The paper doesn't fully explore whether using higher-level actions (as shown in the LLM Agent approach in Figure 4) could be more effective than the sentence-level splitting used in the main experiments.

**Questions For Authors:**

1. Table 1 shows that PGTS significantly underperforms MCTS on complex tasks like GPQA (e.g., 27.78% vs. 34.85% accuracy for LLaMA3.1-8B). Given that one major advantage of state-of-the-art LLMs like o1-class models is their performance on precisely these difficult reasoning tasks, does this indicate a fundamental limitation of PGTS in enhancing model performance on the most challenging reasoning problems? If so, what modifications might address this limitation?
2. Have you explored training with substantially more examples, particularly for complex tasks like GPQA where performance lags behind MCTS?
3. How does PGTS compare to MCTS in terms of sample efficiency when both are trained on identical data? The current comparison focuses on inference efficiency, but a direct comparison of how effectively each method utilizes the same training examples would provide important context.
4. The paper uses sentence-level segmentation for reasoning steps, while Figure 4 shows the LLM Agent approach using higher-level structured actions. Have you investigated whether using more semantically meaningful action definitions beyond simple sentence splitting could improve performance?

**Relation To Broader Scientific Literature:**

The key contributions of PGTS relate to several important research directions in the broader scientific literature.

**Theoretical Claims:**

The paper does not contain formal mathematical proofs for theoretical claims that require rigorous verification. The primary theoretical contribution is the formulation of the reasoning process as a Tree Search MDP (TS-MDP), which is presented in Section 2.3.2.

---

> ### Author Rebuttal · Authors · 2025-03-31
>
> We thank the reviewer for their thoughtful and constructive feedback. We address the key points raised below:
>
> ## PGTS Performance on Complex Tasks like GPQA
>
> The reviewer correctly notes that PGTS underperforms MCTS on GPQA (27.78% vs. 34.85%). This performance gap can be attributed to three primary factors:
>
> 1. Task Complexity: GPQA consists of graduate-level questions across diverse domains requiring specialized knowledge, making it particularly challenging for a small policy without extensive world knowledge.
> 2. Limited Training Data: GPQA's diversity and complexity means that our limited training data (250 examples) may not provide sufficient coverage across all domains for the policy to learn effective exploration strategies.
> 3. Task-Specific Reward Design: Our current reward design uses the same structure across all tasks, with the intermediate reward simply being the likelihood of each reasoning step, which may not be optimal for GPQA's unique characteristics.
>
> We believe this is not a fundamental limitation of PGTS but rather an opportunity for task-specific enhancement. We plan to address this in future work through:
>
> * Developing more sophisticated reward mechanisms tailored to complex reasoning tasks
> * Exploring synthetic data generation to improve training diversity and coverage
>
> ## Training with More Examples
>
> The plateau observed in Figure 3 occurs primarily for simpler reasoning tasks where 1,000 examples provide sufficient coverage of reasoning patterns. For complex tasks like GPQA, we expect the learning curve would continue to improve with substantially more training examples, especially if they provide better domain coverage.
>
> GPQA presents a particular challenge due to its limited training data availability (we were restricted to 250 examples) and the exceptional breadth of graduate-level topics it covers. This limitation likely contributes to the performance gap compared to MCTS on this specific benchmark.
>
> To address this challenge in future work, we plan to explore synthetic data generation techniques to expand training diversity.
>
> ## Action Definitions Beyond Sentence Splitting
>
> The reviewer raises an important point about action granularity. We chose sentence-level segmentation for simplicity and consistency across diverse tasks. However, we agree that semantically meaningful action definitions could further improve performance, and our PGTS framework can easily adapt to different action deefinitions.
>
> We plan to explore hierarchical reasoning structures and task-specific action definitions in future work. This would allow PGTS to operate at multiple levels of abstraction, potentially enhancing both efficiency and performance.
>
> ## Clarifications on Algorithm 1
>
> Thank you for noting the undefined symbols in Algorithm 1. We will revise the algorithm description to clearly define:
>
> - H: The entropy of the policy outputs, which is used as regularization to encourage exploration.
> - $\omega$: The entropy regularization coefficient (set to 0.01 in our experiments)
> - $\eta_{\pi}$ and $\eta_{V}$: Learning rates for the policy and value networks (both set to 3e-4)
>
> We will include these definitions in the final version.
>
> ## Ablation on Cost Function Settings
>
> We acknowledge the lack of ablation analysis on the cost function $C(a)$. In our preliminary experiments, we found that performance is relatively robust to cost settings within reasonable ranges, with the relative ordering (backtrack > branch > expand > terminate) being more important than absolute values.
>
> We'd be happy to include a more detailed analysis of cost function sensitivity in the final version.
>
> We appreciate the reviewer's thoughtful comments and will incorporate these suggestions in the final version.

---

> > ### Comment · Reviewer_GYUV · 2025-04-03
> >
> > Thank you for your detailed rebuttal. While I appreciate your explanations regarding PGTS performance on complex tasks like GPQA, I still believe that more experimental analysis is needed. As you mentioned, 250 examples are insufficient for the model to properly learn this type of knowledge. I hope you will further supplement your experiments in the final version.

---

### Decision · Program_Chairs · 2025-05-01

**Decision:**

Accept (poster)

**Comment:**

This paper received four high-quality reviews, with overall scores of 4, 4, 2, and 4.

All reviewers recognize the innovative nature of the approach and its promising empirical results. Even the reviewer who provided a lower rating acknowledged the solid contribution of the work, though they expressed some concerns regarding clarity in presentation and the additional computational costs introduced during the training phase.

The authors' rebuttal adequately addressed most of the concerns raised. Notably, the critical reviewer affirmed that the presented results are indeed publishable following the authors' clarifications.

After evaluating the reviews and authors' responses, the Area Chair agrees that the paper clearly meets the acceptance criteria. However, it is suggested that further clarification and refinement of presentation be included in the final manuscript.